# Combinatorial discovery of microtopographical landscapes that resist biofilm formation through quorum sensing mediated autolubrication

Bio-instructive materials that intrinsically inhibit biofilm formation have significant anti-biofouling potential in industrial and healthcare settings. Since bacterial surface attachment is sensitive to surface topography, we experimentally surveyed 2176 combinatorially generated shapes embossed into polymers using an unbiased screen. This identified microtopographies that, in vitro, reduce colonization by pathogens associated with medical device-related infections by up to 15-fold compared to a flat polymer surface. Machine learning provided design rules, based on generalisable descriptors, for predicting biofilm-resistant microtopographies. On tracking single bacterial cells we observed that the motile behaviour of *Pseudomonas aeruginosa* is markedly different on anti-attachment microtopographies compared with pro-attachment or flat surfaces. Inactivation of Rhl-dependent quorum sensing in *P. aeruginosa* through deletion of *rhlI* or *rhlR* restored biofilm formation on the anti-attachment topographies due to the loss of rhamnolipid biosurfactant production. Exogenous provision of *N*-butanoyl-homoserine lactone to the *rhlI* mutant inhibited biofilm formation, as did genetic complementation of the *rhlI*, *rhlR* or *rhlA* mutants. These data are consistent with confinement-induced anti-adhesive rhamnolipid biosurfactant 'autolubrication'. In a murine foreign body infection model, anti-attachment topographies are refractory to *P. aeruginosa* colonization. Our findings highlight the potential of simple topographical patterning of implanted medical devices for preventing biofilm associated infections.

Bacteria generally grow as individual planktonic cells in suspension or as organized biofilm communities usually attached to a surface[1]. Within biofilms bacteria form aggregates in an extracellular polymeric substances (EPS) matrix that also facilitates surface colonization. Biofilms possess 'emergent self-organizing properties'[2] that distinguish them from free-living bacterial cells and are the most prevalent bacterial lifestyle. Bacteria arriving at a surface through simple settling or via flagellar-mediated motility initially attach reversibly, before transitioning to irreversible attachment, the first committed step in biofilm formation[1]. This is generally followed by bacterial self-assembly into microcolonies then growth and development into mature biofilms where aggregated cells are embedded in a self-generated EPS matrix consisting of exopolysaccharides, proteins, lipids and extracellular DNA (eDNA)[1,3]. Transitioning from a planktonic to a biofilm lifestyle depends on regulatory decisions informed by sophisticated, integrated surface sensing networks. These involve surface appendages

✉ e-mail: morgan.alexander@nottingham.ac.uk; paul.williams@nottingham.ac.uk

(e.g. flagella, type IV pili (T4P) and adhesins) working in conjunction with sensor regulatory, chemosensory, cell-cell communication (quorum sensing (QS)) systems and second messengers such as cyclic diguanylate (c-di-GMP)[3–6].

Bacterial biofilms present diverse, important challenges for animal and human health, food and water safety as well as oil and gas production[7]. Their emergent properties include intercellular co-operation, nutrient capture and sharing, and enhanced tolerance to environmental stresses such as antimicrobials and host immune defences[2]. In a clinical context, bacterial biofilm formation on the surfaces of implanted medical devices leads to increased patient morbidity and mortality and constitutes a global healthcare problem. Therefore, inhibiting bacterial biofilm development on such devices to reduce the risk of chronic, persistent infections is a major unmet medical need[8].

Attempts to prevent such infections have often focused on the incorporation of leachable bactericidal antimicrobials into medical devices[9]. However, the main drawbacks of this strategy include limited long-term efficacy, the tolerance of biofilms towards antimicrobial agents and selection for antimicrobial resistance[9]. Inhibition of biofilm formation is clearly preferable to systemic delivery of antibiotics for prophylaxis or for treatment after infections arise.

Non-leachable biomaterials that instruct bacterial and host cells proximal to implanted medical devices are a promising, novel approach to reducing infection and improving device performance. However, this requires a better mechanistic understanding of bacteria-surface interactions from the perspective of both the bacterium and the nature of the surface. Consequently, large diversity screening approaches have been applied to identify cell-instructive polymers for a range of biological applications[10]. Successful approaches include identification of polymers that stimulate stem cell proliferation and phenotype control in regenerative medicine[11–13], immune cell instruction for implant device control[14], and polymers that intrinsically resist biofilm formation[15]. Using machine learning, strongly predictive relationships linking physicochemical properties of materials to biological responses have subsequently been identified, allowing the design of polymer chemistries with greatly improved biofilm inhibitory properties[16,17].

Despite the vastness of materials space from which useful materials could be selected, regulatory requirements and costs have greatly restricted the number of materials approved for use in implantable devices. Modification of the surface topography of materials provides a very useful way of strongly influencing cellular responses to surfaces, potentially using materials that have already been approved for use in humans. Nano- and micro- topographic cues have been used to influence human stem[18] and immune cell differentiation[19], marine biofouling[20], and bacterial surface colonization[21,22]. Thus, appropriate micro- or nano-topographic surface features could provide a new means of controlling biofilms in many applications, especially those with direct impact on human health[9,22]. However, although surface feature shape, height, and spacing are known to influence cell attachment[21,22], a general understanding of how topography achieves this is lacking. There is therefore a need to determine mechanistically how topography modulates bacterial cellular responses. This knowledge gap hampers rational design of topographical biomaterials[23] that could substantially improve the efficiency of discovery and the efficacy of their bacterial biofilm inhibitory properties.

Topographies that inhibit bacterial attachment and subsequent biofilm formation can broadly be divided into anti-adhesive and bactericidal surfaces[9,24]. Among these, nature-inspired anti-biofouling surfaces including shark skin, insect wings, and plant flowers and leaves have been exploited to design topographical mimics[25,26]. For example, Sharklet AF, which emulates the topography of shark skin scales, reduced the attachment of bacterial pathogens including *Acinetobacter baumannii, Escherichia coli, Klebsiella pneumoniae,*

*Pseudomonas aeruginosa* and *Staphylococcus aureus* in vitro[26]. Furthermore, immunocompromised rats with percutaneously implanted Sharklet AF micropatterned silicone rods infected with *S. aureus* had lower numbers of bacteria colonizing the rods and fewer bacterial cells in both subcutaneous tissues and spleens compared with those implanted with smooth rods[26].

The mechanism(s) by which Sharklet AF biomimetics reduce bacterial attachment raises both physical and biological questions about the nature of these topographical interactions. Although surface roughness is often positively correlated with increased bacterial attachment, in some cases it correlates with reduced attachment[24]. While average surface roughness ($R_a$) and root mean square roughness ($R_{rms}$) are commonly used for characterizing such topographies, roughness alone does not capture spatial information including the geometric details, density, periodicity, symmetry, or the hierarchical arrangement of surface features, many of which play a critical role in bacterial attachment[21,22]. Surfaces with completely different topographies may have similar $R_a$ and $R_{rsm}$ values[24]. Alternatively, the engineered roughness index (ERI), derived from marine algal zoospore settlement studies, combines roughness with the number and size of topological features[27]. However, its predictive accuracy is restricted to a narrow range of feature sizes and it cannot be used to optimize surface topography.

To discover design rules for topographical control of bacterial responses to surfaces (beyond simple geometries or isolated bioinspired topographical designs), we conducted an unbiased screen of combinatorially generated shapes using a high-throughput micro-topographical polymer chip, the TopoChip[12]. This mathematically designed, random combination of basic primitive shapes comprising triangles, circles and rectangles to produce a library of >2000 micro-topographies was previously used to identify surface topographies that induced proliferation or osteogenic differentiation of human mesenchymal stromal cells[12]. Here, we describe the generation of predictive statistical and machine learning models for bacterial pathogen attachment using topographical descriptors that encode geometric elements. We used these models to identify relationships between topographical patterns and their pro- and anti-attachment properties. Random Forest and Multiple Linear Regression with Expectation Maximisation results are reported after finding that the models generated using these were marginally superior to SVM and XGBoost models. The consistency found in results from the different modelling approaches enabled us to identify key feature shape parameters with the highest relevance for bacterial attachment. Notably, we found hit topographies that significantly reduced attachment and subsequent biofilm formation in vitro by pathogens including *P. aeruginosa, Proteus mirabilis, A. baumannii, and S. aureus*. For the small number of polymers considered, we found that the topographic control of the biofilm formation was not significantly influenced by chemistry. However, prior work with a larger library of polymers has also shown a useful relationship between surface chemistry and attachment[28,29].

When visualized using single cell tracking, the early, near surface swimming behaviour of *P. aeruginosa* was markedly different on anti-attachment compared with pro-attachment or flat control surfaces and could be correlated with biofilm resistance. The deletion of key genes involved in motility and surface sensing (*fliC* and *pilA*), and surface signalling (*wspF*), and the constitutive expression of cyclic diguanylate synthase gene (*yedQ*) all failed to promote biofilm formation on anti-attachment micro-topographies. However, inactivation of Rhl-dependent quorum sensing through deletion of *rhlI* or *rhlR* resulted in biofilm formation on the anti-attachment topographies due to the loss of rhamnolipid biosurfactant production. Exogenous addition of *N*-butanoyl-homoserine lactone (C4-HSL) to the *rhlI* mutant inhibited biofilm formation, as did genetic complementation of the Δ*rhlI*, Δ*rhlR* or Δ*rhlA* mutants. These data suggest a mechanism whereby the QS-

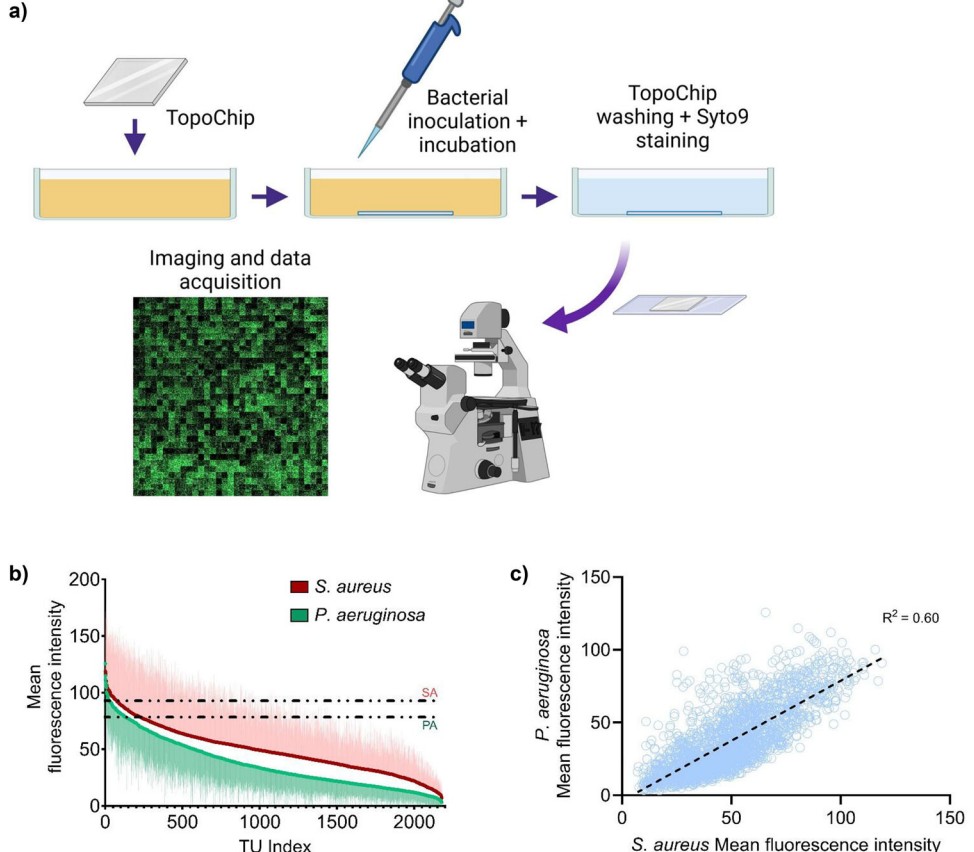

**Fig. 1 | Bacterial attachment to Topochip microtopographies. a** Bacterial attachment assay procedure. Created in BioRender. Bernard, M. (2025) https://BioRender.com/6frm99i. **b** Ranking of topographies according to the mean fluorescence intensity per TopoUnit after *P. aeruginosa* and *S. aureus* attachment to PS TopoChips after 4-h incubation (*n* = 22 and *n* = 6 respectively). The error bars represent one standard deviation. The dotted line corresponds to the mean fluorescence value of the flat surface control for each species. **c** Mean fluorescence intensities from *P. aeruginosa* versus *S. aureus* for each PS TopoUnit. For Fig. 1b, c, the source data are provided as a Source Data file.

controlled production of lubricating rhamnolipids induced via spatial restraints autolubricate the anti-attachment microtopography surfaces and in so doing reduce *P. aeruginosa* surface adhesion preventing irreversible attachment and inhibiting biofilm development. In vivo, a murine foreign body infection model showed that anti-attachment topographies, but not pro-attachment topographies and flat surfaces, were refractory to *P. aeruginosa* colonization. These findings illustrate the strength of unbiased screening to elucidate bacterial cell–surface interactions and provide strategies for the rational design of novel bioinstructive surfaces.

## Results

### Inclusion and ethics statement
This research aligns with the Inclusion & ethical guidelines embraced by Nature Communications.

### Micro-topographical library screen for pro- and anti- attachment surfaces
We used polystyrene (PS) TopoChips consisting of 2176 randomly selected topographical 'features' to sample diverse micro-topographies for their ability to resist bacterial attachment and sub-sequent biofilm formation[12]. These were constructed from three different 'primitive' microscale shapes (circles, triangles and rectangles) that generate different pattern types ranging from those containing large smooth areas to those with angular and stretched elements. The generated topographical feature micropillars had a height of 10 μm and widths ranging from 10 to 28 μm (Supplementary Figs. S1, S2a).

These features were arrayed as 290 × 290 μm TopoUnits surrounded by 40 μm tall walls in a 66 × 66 array that also contained duplicate TopoUnits for each topography and four flat control surfaces (Supplementary Figs. S1a, S2a).

To ensure that the bacteria-material interactions observed were specifically dependent on surface topography, PS TopoChips were subjected to time-of-flight secondary ion mass spectrometry (ToF-SIMS) and X-ray photoelectron spectroscopy (XPS) for quantitative elemental analysis. The results (Supplementary Fig. S3a, b and Supplementary Text S2) confirmed that the surfaces used for screening had uniform chemistries that were suitable for investigating bacterial responses to micro-topographies.

Bacterial attachment and biofilm formation by representative Gram-negative (*P. aeruginosa*) and Gram-positive (*S. aureus*) bacterial pathogens on the TopoUnits is illustrated in Fig. 1a. A 4 h incubation time and static conditions at 37 °C provided a sufficiently stringent assay for the identification of topographies with low and high initial bacterial cell attachment (Fig. 1b and Supplementary Fig. S4). Quantification of *P. aeruginosa* attachment to each TopoUnit on the PS TopoChip (*n* = 22) via fluorescence intensity measurements revealed a diversity of attachment values, plotted in rank order in Fig. 1b. For *S. aureus*, a similarly broad range of attachment was observed from the fluorescence intensity measurements, also rank ordered (Fig. 1b). The attachment of these two pathogens to all TopoUnits was well correlated, as shown in Fig. 1c. This suggested a common response for two different bacterial species, one flagellated and motile (*P. aeruginosa*) and the other non-flagellated (*S. aureus*). In addition, most micro-

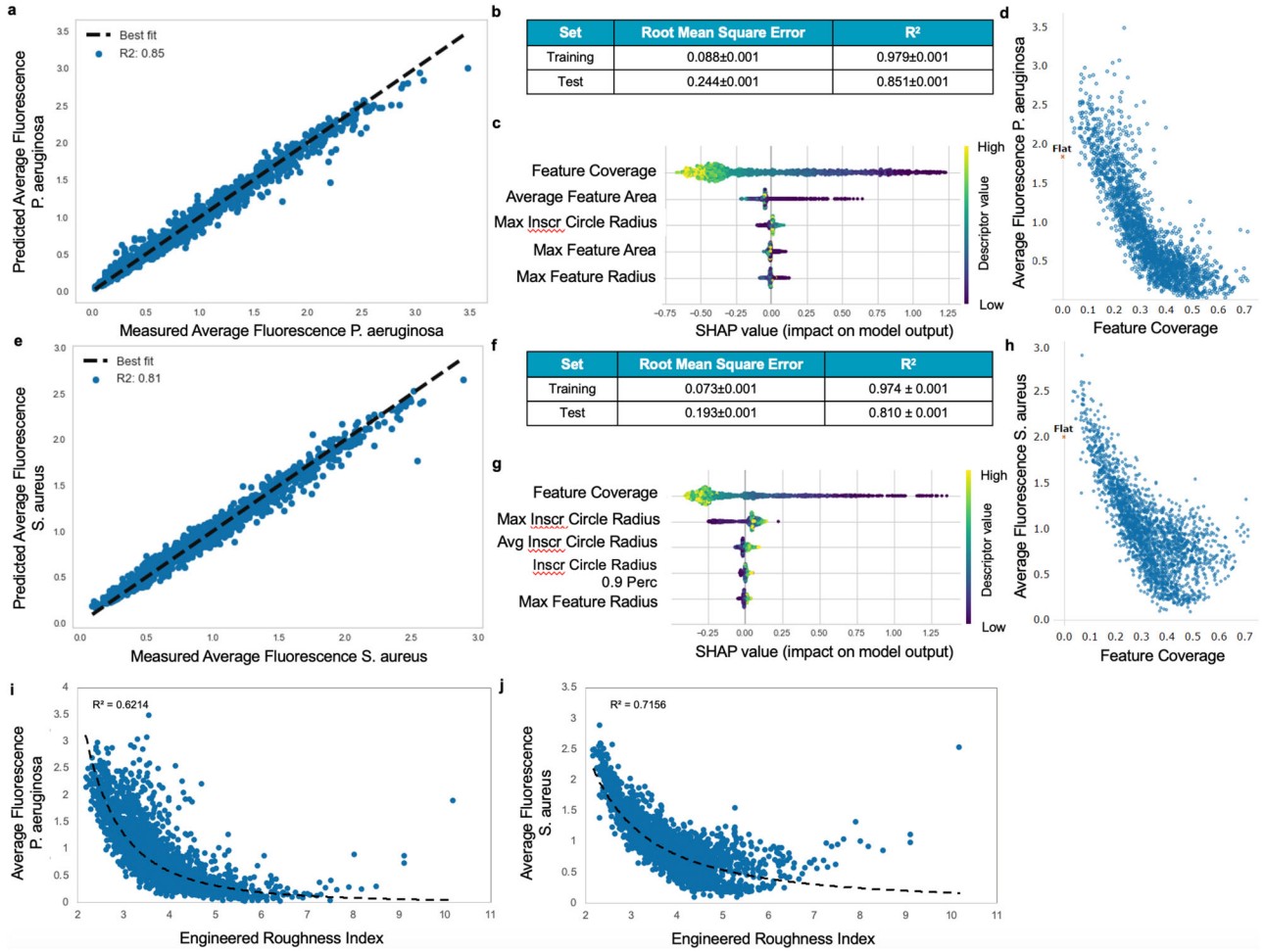

**Fig. 2 | Machine learning modelling results for pathogen attachment using Random Forest.** **a** Plot of the predicted versus measured average attachment values for one run of the *P. aeruginosa* test set, **b** Regression model performance metric results for *P. aeruginosa* training and test sets, **c** *P. aeruginosa* descriptors ranked by their importance from top to bottom and how their value (high = yellow or low = purple) impacts on the model (positive or negative impact), **d** Feature coverage descriptor distribution against attachment for *P. aeruginosa*, **e** Scatter plot of the measured against predicted average attachment values for one run of the *S. aureus* test set, **f** Regression model performance metric results for the *S.* *aureus* training and test sets, **g** *S. aureus* descriptor importance and their impact on model output. The topographical descriptors found to be most important for bacterial attachment are the inscribed circle which relates to the space between primitives, the total area covered by features (feature coverage) and the maximum feature radius, **h** Feature coverage descriptor distribution against attachment for *S. aureus*, **i** Comparison between P. *aeruginosa* attachment and the engineered roughness index (ERI) calculated for the topographies investigated, **j** Comparison between S. *aureus* attachment and the roughness index calculated for the topographies. The source data are provided as a Source Data file.

patterns on the PS TopoChip showed reduced bacterial attachment compared to the flat control, especially for *S. aureus* where only ~4% of the topographies exhibited increased cell attachment (Fig. 1b). These findings strongly suggested that, by modifying surfaces with well-defined micro-topographies, cell attachment and subsequent biofilm development could be inhibited. By visual inspection of the high and low-adhering topographies, fewer bacterial cells were observed attached to topographies with small gaps between the TopoUnit features (see Supplementary Fig. S2a for examples of anti- and pro-attachment TopoUnits).

## Microtopographical design rules for bacterial attachment

To identify the most relevant topographical elements in the TopoUnits from this rich library we employed machine learning (ML) to extract generalisable design rules. Mathematical descriptors for the topographical shapes were generated using the cell shape analysis functions of the widely used CellProfiler and ImageJ software packages. 242 shape descriptors were obtained from the analysis of the TopoUnit designs as described by Vassey et al.[19]. Those with Pearson correlations above 0.85 were removed to eliminate co-variant descriptors, resulting

in 68 that were used to train models of *P. aeruginosa* and *S. aureus* attachment. The topographical descriptors used for modelling are listed in Supplementary Table S1, and images of selected TopoUnits shown in Supplementary Fig. S2a. The Random Forest ML algorithm was used to map the TopoUnit topographic descriptors to the bacterial attachment, resulting in high coefficients of determination ($R^2$) and low root mean square (RMS) errors (Supplementary Text S1). Shapley Additive Explanations (SHAP) were used to identify the descriptors with the largest contributions to the Random Forest models (Supplementary Text S1).

The performance of the *P. aeruginosa* attachment model is shown in Fig. 2a, b, and for *S. aureus* in Fig. 2e, f. The contributions of the five descriptors found to be the most important for each model are shown in Fig. 2c for *P. aeruginosa* and in Fig. 2g for *S. aureus*. The ML model predicted the observed attachment values for topographies in the test sets for both bacterial species with high efficacy: $R^2 = 0.851 \pm 0.001$ for *P. aeruginosa* average fluorescence and $R^2 = 0.810 \pm 0.001$ for *S. aureus* average fluorescence (Fig. 2b, f). The models were trained using the most significant numerical topographical descriptors selected by SHAP. The most important of these were the total area of the

topographical features (feature coverage), the size of gaps between replicates of topographical features on the TopoChip (defined using the radii of largest inscribed circles that fit into the gaps, see Supplementary Fig. S1b), and the maximum radius of the features (defined as the largest feature of each microtopography). The model showed that the topographical feature area was inversely correlated with the bacterial attachment as shown by both the pattern feature coverage and the maximum feature area within the unit cell in which the features are arranged.

To extract design rules from the ML model, we plotted each of the most relevant topographical descriptors against attachment in Supplementary Fig. S5. These graphs showed that the lowest *P. aeruginosa* attachment was associated with topographies with feature coverage >0.5 and maximum inscribed circle radius ≤3 μm, although unique solutions were not found for the maximum feature radius descriptor alone. For *S. aureus*, the lowest attachment was associated with feature coverage >0.4 and maximum inscribed circle radius ≤3 μm. The strong correlation of low attachment with the proportion of surface covered by the topography and its radius, together with the negative correlation with the size of the gaps between replicate topographies (maximum inscribed circle radius), are very useful for guiding the design of other low attachment topographical biomaterials. That is, these rules can be used to develop attachment-resistant surfaces primarily by optimizing the size of the topographical features (feature coverage and maximum feature radius) and the gaps between them (maximum inscribed circle). To investigate the effect of only using the most dominant descriptor, Feature Coverage, we plotted bacterial fluorescence, versus Feature Coverage for both *P. aeruginosa* (Fig. 2d) and S. *aureus* (Fig. 2h) for all topographies, inserting the flat comparator at Feature Coverage = 0. It is apparent that (a) the minimal bacterial colonisation data occurs for designs with Feature Coverages between 0.4 and 0.6 for both pathogens, and (b) that this descriptor alone fits the data with significantly less certainty that the model in Fig. 2a, e.

An 'Engineered Roughness Index' (ERI), calculated from micro-topography dimensions, was previously proposed by Schumacher et al.[20] to characterize the reduction in algal zoospore settlement on 6 different cuboid patterns and Sharklet AF using a smooth surface as a control. They reported that the best performing microtopography was the bioinspired Sharklet AF. This Sharklet microtopography also reduced early bacterial colonization and biofilm formation by bacterial pathogens including *P. aeruginosa* and *S. aureus*[26].

The ERI is related to the size, geometry, and spatial arrangement of the topographical features[20]. These factors are captured by Wenzel's roughness factor ($r$), depressed surface fraction ($f_D$), and degree of freedom for movement (of zoospores in the cited paper[20]) ($df$). Wenzel's roughness factor is the ratio of the exposed surface area to the projected planar surface area. $f_D$ is the ratio of the surface area between features and the projected planar surface area. $df$ describes to the tortuosity of the surface – the ability of a zoospore (or bacterium in the current case) to follow gaps or channels between features on the topographical surfaces. Notably, these elements are similar to the most relevant descriptors in the ML model trained on our topography library, i.e. parameters representing the area of features and the depressed areas between them. We calculated the ERI for all our topographies and plotted the relationship with bacterial attachment in Fig. 2i, j for *P. aeruginosa* and S. *aureus* respectively. Clearly, the trends we observed were consistent with the hypothesis of Schumacher et al.[20] that topographical patterns with higher roughness reduce microbial attachment and that the relationship is non-linear. However, the lower $R^2$ values of models using ERI as a descriptor have lower prediction accuracy than those we derived using topographical descriptors from CellProfiler (Fig. 2a, e). This is likely due to the ML model identifying more precise measures of surface properties for the topographies in the library.

## Micro-topographical interactions are independent of pathogen, growth environment and materials chemistry

Our experiments identified TopoUnits 881, 685, 632 and 1497 as the best performing, exhibiting 5–20 fold reductions in bacterial attachment relative to flat surfaces and pro-attachment controls (336, 697, 1527 and 1556) (Supplementary Fig. S2a). They were selected for further study as anti-attachment topographies. Figure 3 shows representative images (Fig. 3a) and quantitative data (Fig. 3b) for selected TopoUnits for *P. aeruginosa* and *S. aureus*. We also show the responses of two additional human pathogens, *Pr. mirabilis* (flagellated) and *A. baumannii* (non-flagellated) grown under static conditions for 4 h. The data show that all four pathogens colonized the pro- and flat surfaces but not the anti-attachment topographies (Supplementary Fig. S2b and Fig. 3).

We next investigated the performance of the TopoUnits with respect to growth environment for *P. aeruginosa*. The contribution of gravity to bacterial colonization was assessed by inverting the culture set up. The key topographies (Supplementary Fig. S2a) maintained their *P. aeruginosa* pro- and anti-attachment properties independent of orientation and the presence of flow conditions (Fig. 4a, b). No obvious differences in the morphology of *P. aeruginosa* single cells were apparent on flat, pro- attachment (TopoUnit 697) or anti-attachment (TopoUnit 881) micro-topographies (Supplementary Fig. S6). The attachment of *P. aeruginosa* to two other polymers (polyurethane (PU) and cyclic olefin copolymer used clinically to fabricate medical devices) was also assessed after embossing with the anti- and pro-biofilm topographies (Fig. 4a, b). The most effective low attachment micro-topographies for PS were also the most effective for the other two polymers. This indicates that surface topography influences bacterial attachment in the same way for these three materials with differing surface chemistries (Fig. 4a, b).

To determine whether *P. aeruginosa* and *S. aureus* form biofilms on pro- and anti-attachment topographies after an extended incubation time, they were grown on PS TopoUnits for 24 h (Fig. 4c, d). The anti-attachment TopoUnits 881 and 685 showed more than 15-fold lower *P. aeruginosa* biofilm biomass compared with the flat surface (Fig. 4d) (two-way ANOVA, $p < 0.0001$). Mature biofilms incorporating interconnecting bacterial aggregates were clearly observed on the pro-attachment TopoUnits 697 and 336, as well as on the flat control, while the few cells present on the anti-attachment TopoUnits 881 and 685 were far more dispersed. Similar results were also obtained for *S. aureus* (Fig. 4c, d) where a 5-fold (two-way ANOVA, $p < 0.0001$) average reduction of surface growth on the two anti-attachment TopoUnits were observed after 24 h incubation compared with the flat control surface. Collectively, these data show that the 4 h attachment data was a good indicator of subsequent biofilm resistance.

## Exploration of micro-topographies by *P. aeruginosa*

The success of bacterial surface exploration depends on the interplay between surface sensing and the local environment including surface physicochemical properties and architecture[22]. To gain insights into the molecular basis by which anti-attachment TopoUnits such as 881 prevent biofilm formation, we focused on *P. aeruginosa*. This was because of the extensive knowledge on the mechanisms used by this Gram-negative bacterium to interrogate surfaces[1,3–6]. We first considered whether the lack of *P. aeruginosa* colonisation of anti-attachment topographies was due to motility-mediated surface avoidance or detachment following initial attachment. This would result in the failure of the bacterial cells to become irreversibly attached, the first committed step in biofilm development[1]. To investigate the motile behaviour of *P. aeruginosa* on the micro-topographies, single cells interacting with flat, pro- and anti-attachment TopoUnits were tracked using video differential interference microscopy (DIC) within 2 min of inoculation, and after 2 h and 4 h incubation. Bacterial cells were categorized as either swimming or stationary.

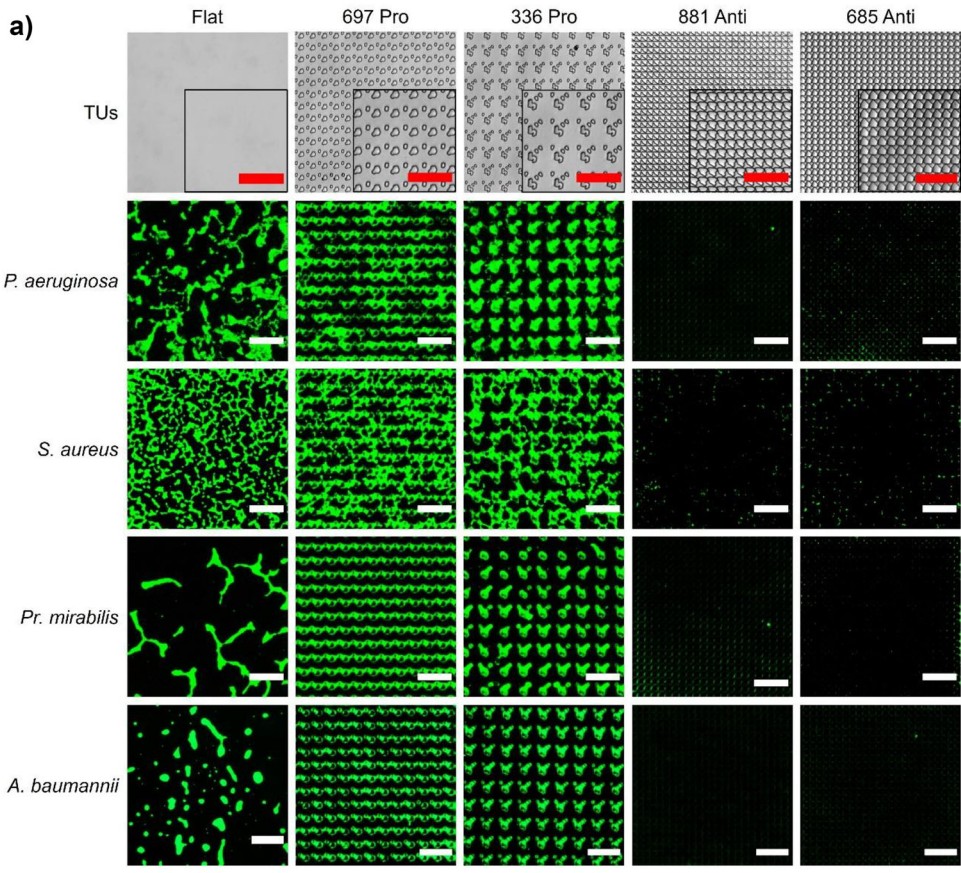

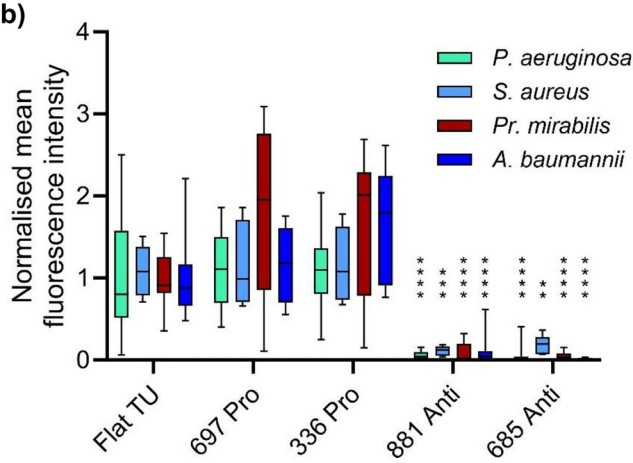

**Fig. 3 | Attachment of bacterial pathogens to flat, pro- (Topo-Units 697 and 336) and anti-attachment (881 and 685) PS. a** Representative images of *P. aeruginosa*, *S. aureus*, *Pr. mirabilis* and *A. baumannii* attachment (bright field images of the TopoUnits shown in top row) after 4 h incubation under static conditions. Scale bar: 50 μm. **b** Quantification of normalised mean fluorescence intensity of bacterial cells stained with Syto9 and grown on the same topographies (*P. aeruginosa* n = 22 TopoUnits, *S. aureus* n = 6 TopoUnits, *Pr. mirabilis* n = 12 TopoUnits and *A. baumannii* n = 16 TopoUnits). Data shown in boxes extend from the 25th to 75th percentiles and lines in boxes correspond to the median values. Whiskers go down to the smallest and up to the largest values. Statistical analysis was done using a two-way ANOVA with Dunnett's multiple comparisons test (*$p < 0.05$; **$p < 0.01$; ***$p < 0.001$; **** $p < 0.0001$). *P*-values for 881 and 685 topographies are: *P. aeruginosa* = 1.28E−12 and 7.05E−13, *S. aureus* = 0.00032 and 0.0011, *Pr. mirabilis* = 9.25E−07 and 3.22E−07, *A. baumannii* = 2.72E−09 and 6.37E−11. The source data are provided as a Source Data file.

The individual swimming cells tracked immediately following exposure to anti-attachment TopoUnit 881 showed confinement within the wider (2 μm) channels (Fig. 5a). The direction of movement within the channels was primarily linear (Fig. 5a, d; Supplementary Movie 1). This behaviour is reminiscent of *Vibrio fischeri* cells entering a confined space and responding by straightening their swimming paths to facilitate escape from confinement[30]. Such confinement and directionality were not observed for *P. aeruginosa* cells at this early time point when moving over a pro-attachment TopoUnit 697 or on a flat PS surface (Fig. 5b–d). Typically, there was greater free

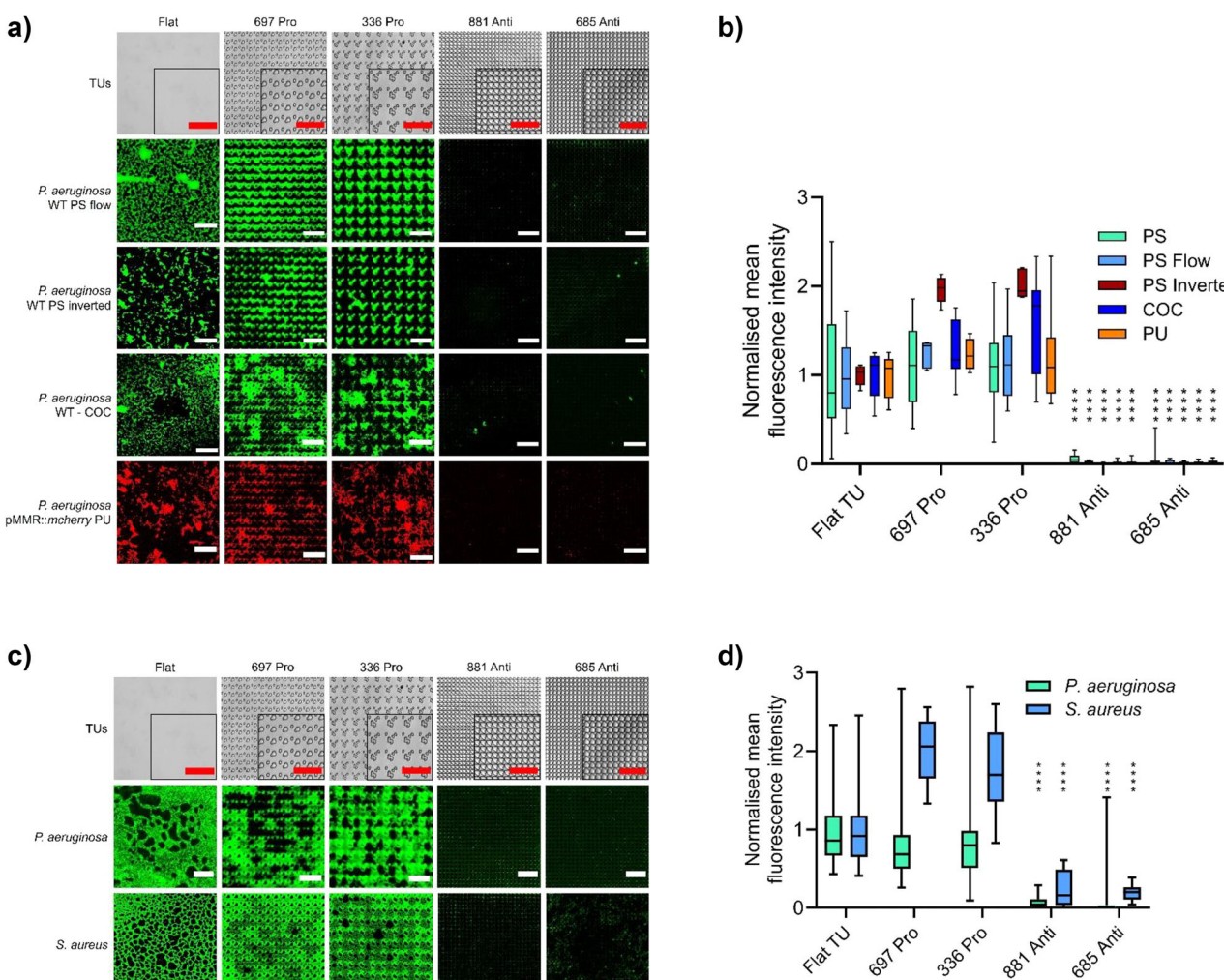

**Fig. 4 | Bacterial attachment comparing different materials, inversion and flow.** **a** Representative images of *P. aeruginosa* wildtype stained with Syto9 fluorescent dye (green) or *P. aeruginosa* wild type transformed with mCherry (red; on PU only) grown for 4 h on flat, pro- or anti-attachment TopoUnits (bright field images shown in top row) moulded from PS, PU or cyclic olefin copolymer (COC). *P. aeruginosa* wildtype attachment on upside down oriented TopoUnits (inverted) and under flow conditions is shown for PS TopoUnits. Scale bar: 50 µm. **b** Quantification of normalised mean fluorescence intensity of wildtype *P. aeruginosa* incubated under conditions described above (PS: *n* = 22 TopoUnits, PS Flow: *n* = 4 TopoUnits, PS Inverted: *n* = 5 TopoUnits, COC: *n* = 12 TopoUnits and PU: *n* = 8 TopoUnits). Statistical analysis was done using a two-way ANOVA with Dunnett's multiple comparisons test (*$p < 0.05$; **$p < 0.01$; ***$p < 0.001$; ****$p < 0.0001$). *P*-values for 881 and 685 topographies are: PS < 1.00E−15 and <1.00E−15, PS Flow = 2.40E−06 and 6.23E

−9, PS Inverted = 7.84E−06 and 2.41E−05, COC = 3.37E−13 and 7.46E−13, PU < 1.00E−15 and <1.00E−15. **c** Bacterial biofilm assessment: Representative images of *P. aeruginosa* and *S. aureus* biofilms on flat, pro and anti-attachment PS topographies after 24 h incubation under static conditions. Scale bar: 50 µm. **d** Quantification of mean normalised fluorescence intensity for wildtype *P. aeruginosa* and *S. aureus* biofilms shown in **c**. *P. aeruginosa* *n* = 32 TopoUnits, *S. aureus* *n* = 24 TopoUnits. Data shown in boxes extend from the 25th to 75th percentiles and lines in boxes correspond to the median values. Whiskers go down to the smallest and up to the largest values. Statistical analysis was done using a two-way ANOVA with Dunnett's multiple comparisons test (* $p < 0.05$; ** $p < 0.01$; *** $p < 0.001$; **** $p < 0.0001$). *P*-values for 881 and 685 topographies are: *P. aeruginosa* = 2.22E−13 and 2.22E−13, *S. aureus* = 5.29E−10 and 1.08E−10. For Fig. 4a, d, the source data are provided as a Source Data file.

movement over the surface and between and over the much more widely spaced feature pillars, with early settlement of *P. aeruginosa* cells close to the pillar bases (Supplementary Movies 2 and 3). This finding was consistent with the fluorescent images observed after 4 h shown for *P. aeruginosa* in Figs. 3a and 4a where the pro-attachment TopoUnit feature pillars were more highly colonized with fluorescent cells compared to the gaps between them. Interestingly, the proportion of stationary bacteria was significantly lower for the anti-attachment TopoUnit 881 at the early 2 min incubation time compared to both the flat control and 697 TopoUnit (Fig. 5e). These findings correlated with the amount of subsequent attachment after 4 h incubation (Fig. 3a, b for *P. aeruginosa*). However, after 2 h and 4 h incubation, most of the cells were stationary, irrespective of the surface (Supplementary Fig. S7). The differential motility of *P.*

*aeruginosa* observed after 2 min incubation suggested that biofilm development on a specific microtopography could be predicted from the tracked behaviour of the bacteria at this early time point. On TopoUnit 881, after both 2 and 4 h incubation, stationary bacteria were observed localized within the crevices between the feature pillars (Supplementary Fig. S7).

## Surface sensing and microtopography colonization

In *P. aeruginosa*, cell surface appendages essential for flagellar-mediated swimming and T4P-mediated twitching motility are also involved in surface sensing[31–33]. Activation of both Pil-Chp and Wsp chemosensory systems results in increased production of c-diGMP that subsequently drives the production of extracellular matrix and biofilm development[34–37].

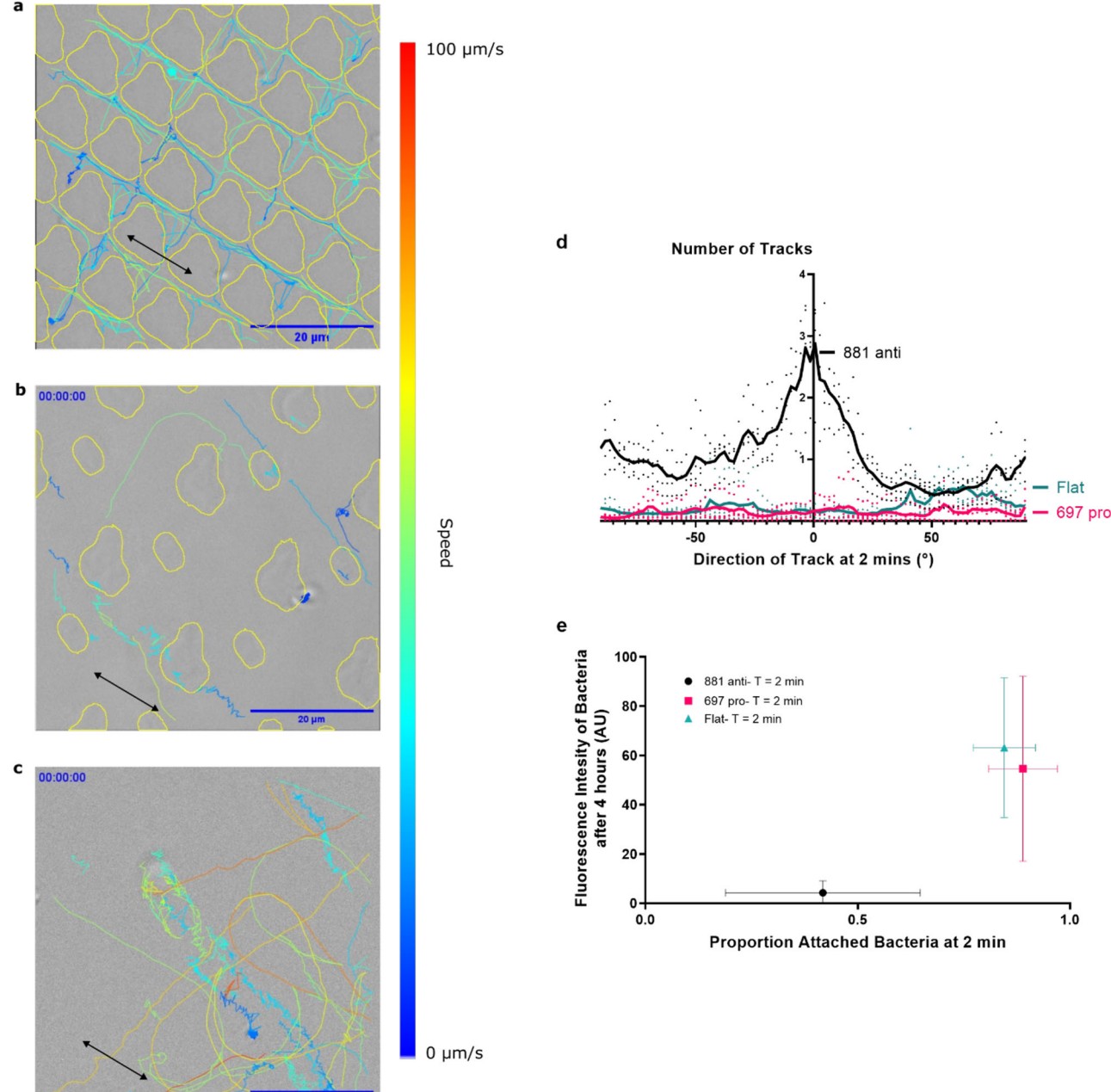

**Fig. 5 | Differential interference contrast (DIC) microscopy of *P. aeruginosa* cells within 2 min of inoculation, tracked over 20 s (frames captured at 20 ms intervals) for 2 min and converted into lines representing the path of an individual cell.** Overlays of individual cell tracks on **a** anti-attachment TopoUnit 881, **b** pro-attachment TopoUnit 697 and **c** flat control on a single frame with yellow lines indicating the positions of the subtracted topographical features (scale bars, 20 μm). Track speed is indicated by line colour. Black double-headed arrows indicate the directionality of the cells relative to the 2 μm wide channels in TopoUnit

881. **d** Number of tracks and their directionality after 2 min incubation on 881 (black), 697 (magenta) and flat control (green) surfaces respectively. **e** The proportion of stationary surface associated cells after 2 min correlates with subsequent bacterial attachment (fluorescence intensity; AU, arbitrary units) after 4 h incubation. Data shown are mean ± SD. *n* = 739, 813 and 809 single bacterial cells respectively tracked on flat, pro-attachment (TopoUnit 697) and anti-attachment (TopoUnit 881) surfaces. For Fig. 5d, e, the source data are provided as a Source Data file.

We investigated the contribution of surface sensing to the differential attachment of *P. aeruginosa* to diverse micro-topographies. We first examined the interactions of *P. aeruginosa* wild-type and the isogenic flagellin (Δ*fliC*) and T4P (Δ*pilA*) deletion mutants with the selected TopoUnits to determine whether the loss of swimming or twitching motility impacted on colonization. Figure 6a, b shows that, like the wildtype, neither Δ*pilA* nor Δ*fliC* colonized the anti-attachment TopoUnit. However, both mutants showed reduced levels of attachment to both flat and the pro-attachment TopoUnit (Fig. 6b). Thus, neither the

loss of flagellum nor T4P rendered *P. aeruginosa* 'blind' to surface microtopography. Although neither surface appendage was essential for colonization of the pro-attachment TopoUnit, the lack of T4P resulted in some microcolony segregation, where cells were more prone to colonize the micro-topographical features (Fig. 6a). These results, combined with single cell tracking of *P. aeruginosa* on selected topographies, suggested that the differences in attachment to pro- and anti- TopoUnits were not simply related to inability of the bacterial cells to move to certain topographical areas but to differences in surface sensing.

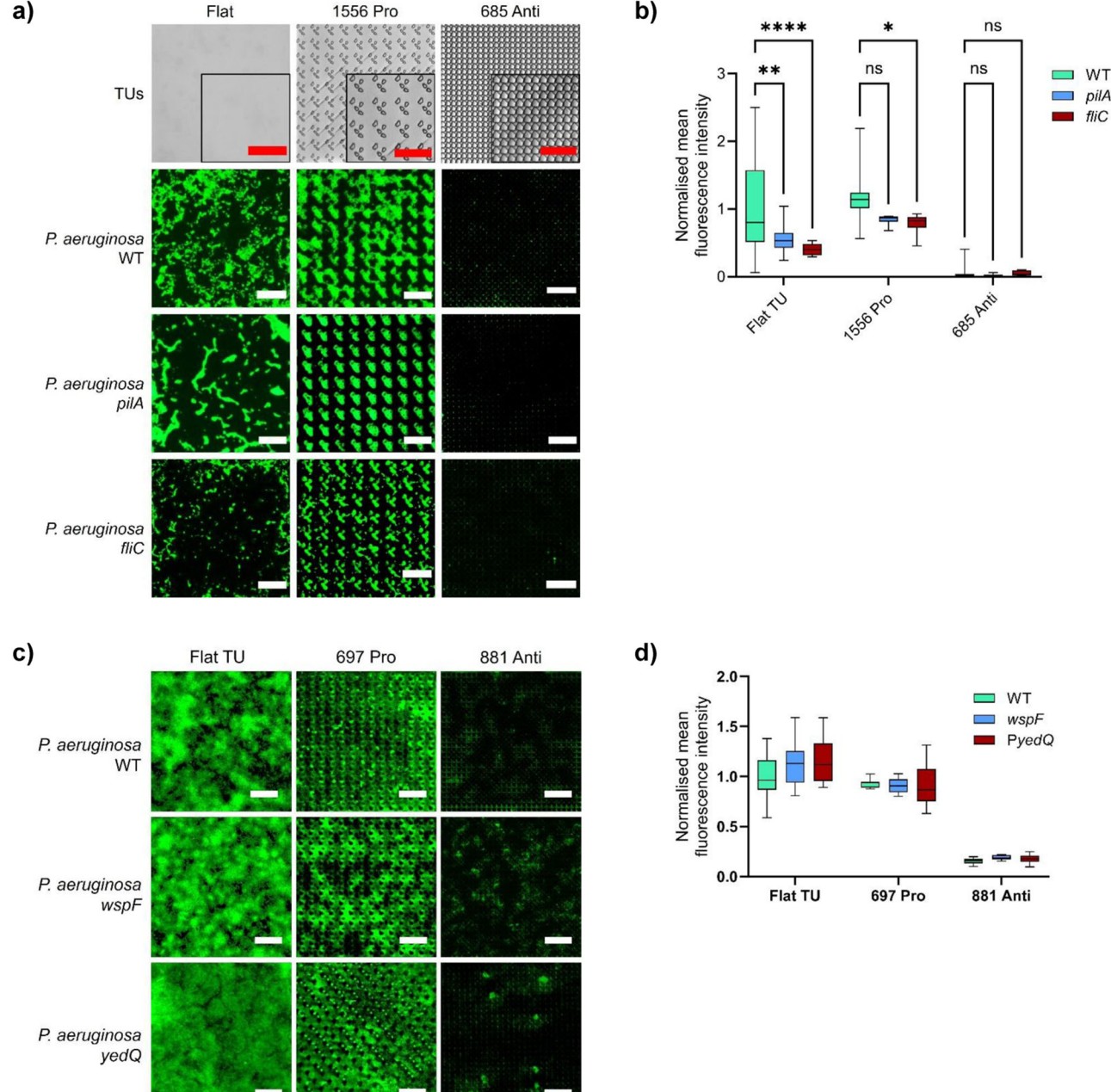

**Fig. 6 | Mutation of flagella, T4P, *wsp* signalling or overexpression of the diguanylate cyclase *yedQ* do not facilitate colonization of anti-attachment TopoUnits.** Representative fluorescent images **a** and normalized mean fluorescence intensities **b** of *P. aeruginosa* wildtype, Δ*pilA* and Δ*fliC* cells attached to flat, pro and anti-attachment PS topographies after 4 h incubation (WT: *n* = 22 TopoUnits, *pilA*: *n* = 6 TopoUnits, *fliC*: *n* = 9 TopoUnits). Statistical analysis was done using a two-way ANOVA with Dunnett's multiple comparisons test (*$p < 0.05$; **$p < 0.01$; ***$p < 0.001$; ****$p < 0.0001$; ns not significant). P-values WT vs *pilA*: Flat 0.0015, 1556 0.17 and 685 0.98. P-values WT vs *fliC*: Flat 1.72E-05, 1556 0.041 and

685 0.99. Scale bar: 50 µm. Representative images **c** and normalized mean fluorescence intensity **d** of *P. aeruginosa* wildtype, Δ*wspF* and the wild type expressing *yedQ* (p*yedQ*) on flat (*n* = 16 TopoUnits), pro (*n* = 10 TopoUnits) and anti-attachment (*n* = 7 TopoUnits) surfaces after 24 h incubation under static conditions. Data shown in boxes extend from the 25th to 75th percentiles and lines in boxes correspond to the median values. Whiskers go down to the smallest and up to the largest values. Scale bar: 50 µm. For Fig. 6b, d, the source data are provided as a Source Data file.

To determine whether the reduced attachment to microtopographies such as 881 was related to a lack of Wsp signalling in *P. aeruginosa*, the attachment of a *wspF* deletion mutant[34] on selected topographies was quantified. Inactivation of *wspF* locks the *wsp* system into a constitutively active state, stimulating c-di-GMP production and driving EPS production and biofilm development[34,37]. Figure 6c, d shows that, deletion of *wspF* did not enhance biofilm formation on the anti-attachment TopoUnit 881. Since *P. aeruginosa* has multiple

chemosensory and c-di-GMP signalling systems[5,6], we constitutively overproduced intracellular c-di-GMP by introducing the *E. coli* c-di-GMP synthase, YedQ[38] into *P. aeruginosa*. In common with the *wspF* mutant, overexpression of *yedQ* failed to drive biofilm formation on the anti-attachment topography 881 (Fig. 6c, d). These results suggests that surface mechanical cues, Wsp signalling, or differences in c-di-GMP levels cannot explain the anti-attachment properties of specific micro-topographies.

### QS-dependent rhamnolipid production prevents biofilm formation on anti-attachment micro-topographies

*P. aeruginosa* produces rhamnolipid biosurfactants that have diverse biological properties and contribute to surface migration via twitching motility, QS signal molecule solubilization, biofilm maturation and dispersal[39,40]. They also act as surface lubricating anti-adhesives[41] and antimicrobials and are regulated via QS, where the *rhl* QS system is particularly important[42,43]. Since rhamnolipids have anti-adhesive properties, we investigated whether exogenous provision of purified rhamnolipids or abolition of their biosynthesis by mutation of the *rhlA* gene influenced biofilm formation on the micro-topographies. Quantitative data and representative confocal images are shown in Fig. 7. Addition of exogenous rhamnolipids to the parent *P. aeruginosa* strain inoculum significantly reduced 24 h biofilm formation on both the pro-attachment 697 TopoUnit and the flat control (Fig. 7a, b). In contrast, the *rhlA* mutant formed biofilms on anti-attachment TopoUnit 881 as well as the pro-attachment topography 697 and the flat control (Fig. 7a, b). Genetic complementation of the ΔrhlA mutant restored rhamnolipid production (Supplementary Fig. S8) and inhibited biofilm formation on 881 (Fig. 7a, b). Exogenous rhamnolipids substantially reduced biofilm formation by the *rhlA* mutant on TopoUnit 697 and the flat surface (Fig. 7a, b).

Rhamnolipid production is regulated by QS, notably via the transcriptional regulator RhlR that is activated by the RhlI-generated QS signal molecule, *N*-butanoyl-homoserine lactone (C4-HSL)[43,44]. Thus, we studied the biofilm phenotype of ΔrhlI and ΔrhlR mutants that are respectively unable to synthesize rhamnolipids because C4-HSL signal molecule biosynthesis is abolished or because they lack the transcriptional activator, RhlR (Supplementary Fig. S8). Figure 7c, d shows that the ΔrhlI mutant forms a biofilm on anti-attachment TopoUnit 881 that could be reversed by genetic complementation with plasmid-borne *rhlI* or provision of exogenous C4-HSL. Consistent with these data, deletion of the *rhlR* gene also resulted in biofilm formation on the TopoUnit 881 that was also reversed by genetic complementation (Fig. 7c, d).

As single cell tracking on the anti-attachment 881 TopoUnit revealed the cumulative trapping of bacteria in the crevices between the feature pillars, this suggested that localized accumulation of bacterial cells in this confined niche results in the early induction of QS (Fig. 5). This would in turn lead to premature rhamnolipid production and explain the inability of *P. aeruginosa* to adhere to the anti-attachment micro-topographies.

### Anti-attachment topographies exhibit biofilm inhibitory properties in a murine foreign body infection model

Prior to conducting in vivo experiments, it was important to determine whether surface conditioning by serum proteins influenced the interaction of bacterial cells with pro- and anti-attachment TopoUnits. Hence, we grew *P. aeruginosa* in tryptic soy broth (TSB) with or without human serum (10 %v/v) and compared protein adsorption to flat, pro- and anti-attachment TopoUnits after incubation at 37 °C for 4 h. Surface chemical analysis in Supplementary Fig. S3 shows uniform deposition of serum protein on the pro- and anti- attachment TopoUnits. Significantly, the relative attachment behaviours of *P. aeruginosa* on flat, pro- and anti- attachment TopoUnits were not affected by serum protein deposition (Supplementary Fig. S9).

To determine whether the in vitro biofilm inhibiting properties of the bacterial attachment micro-topographies were retained in vivo, either scaled-up PU (7 mm × 2 mm) flats or TopoUnits (697 and 881) were inserted subcutaneously in BALB/c mice (one implant per mouse; $n = 8$ for each TopoUnit). PU is a common biomaterial used for fabricating medical devices and was used since it is sufficiently flexible not to irritate the mice after implantation. After insertion, the mice were allowed to recover for 4 days prior to inoculation with either *P. aeruginosa* (tagged with the fluorescent protein E2Crimson;

Supplementary Table S4) or phosphate-buffered saline (PBS; uninfected control). This method enabled us to mimic implantation and colonization of a medical device and to evaluate the combined impact of host and microtopography on the course of infection. After 8 days in the mice, the implants and surrounding tissues were recovered, fixed and sectioned. Retention of micro topographical feature integrity during the experiment was confirmed by environmental scanning electron microscopy (Supplementary Fig. S10a).

Infected mice implanted with TopoUnit 881 showed significantly lower *P. aeruginosa* pcE2C colonization compared with those receiving flat or TopoUnit 697 as determined using quantitative confocal fluorescence microscopy (Fig. 8a, b). Antibody-based staining of *P. aeruginosa* on the explanted 881 and 697 TopoUnits confirmed these findings (Fig. 8c). Since the TopoChips were implanted 4 days prior to infection, it is possible that host cell colonization of the implants contributed to the reduction in bacterial attachment, in essence winning the 'race for the surface'. However, Fig. 8c clearly shows that for TopoUnit 697, both host cells and bacteria maintained access to the microtopography. Furthermore, an asymmetric host response to the TopoUnits, illustrated by staining of a representative PU TopoUnit 881 recovered from an un-infected (PBS-inoculated) animal is shown in Fig. 8d. Here, the implant was removed from the mouse tissues during sectioning, leaving a void shown as a black area in each image. Host responses to the micro-topographically patterned (top) and flat sides (bottom) were very different. Greater host cell migration into the TopoUnit side of the implant (indicated by the blue DAPI-stained nuclei) compared with the flat underside is clearly apparent (Fig. 8d). These data demonstrate that Topounit 881 maintained its colonization resistance properties in vivo.

## Discussion

Microscale surface topographic features may support or inhibit bacterial attachment and subsequent biofilm formation, depending on their shape, size, and density. However, the interplay between bacterial cells and topographical landscapes is relatively poorly understood. It involves not only the physicochemical nature of the surfaces, but also environmental sensing by both individual bacterial cells and populations that together determine their subsequent behaviour[21,22,24,45]. Attachment of bacteria to micro-topographies is known to be dependent on surface wettability, ordering and segregation of cells, chemical conditioning of the surface, and hydrodynamics in the presence of flow[22]. While the physical mechanisms driving bacterial attachment differ for topographical scales, most studies have focused on a limited number of topographical designs[21,24,26,46]. This approach has hampered development of computational models able to predict bacterial responses from materials surface properties. To address these limitations and to discover novel micro-topographies that prevent bacterial attachment and subsequent biofilm formation, we screened TopoChip arrays against two important pathogens, *P. aeruginosa* and *S. aureus*. The results of these screens were used to train ML models that mapped biological outcomes to surface properties, allowing iterative prediction and testing that facilitates optimization of 'hit' topographies. The large number of diverse micro-topographies evaluated and the strong quantitative relationships identified between topographical landscape and attachment for both pathogens, enabled identification of the most relevant topographical features modulating bacterial behaviour. This approach allowed prediction of bacterial attachment from specific surface design criteria rather than from surface roughness, hydrophobicity, or other physical surface properties that did not provide general principles connecting surface topography with bacterial attachment[21,22,24].

Clearly, for biofilm inhibition, surface topographies associated with low pathogen attachment are required. Our ML models have identified the most relevant topographical feature descriptors contributing to bacterial attachment. Features with large negative model

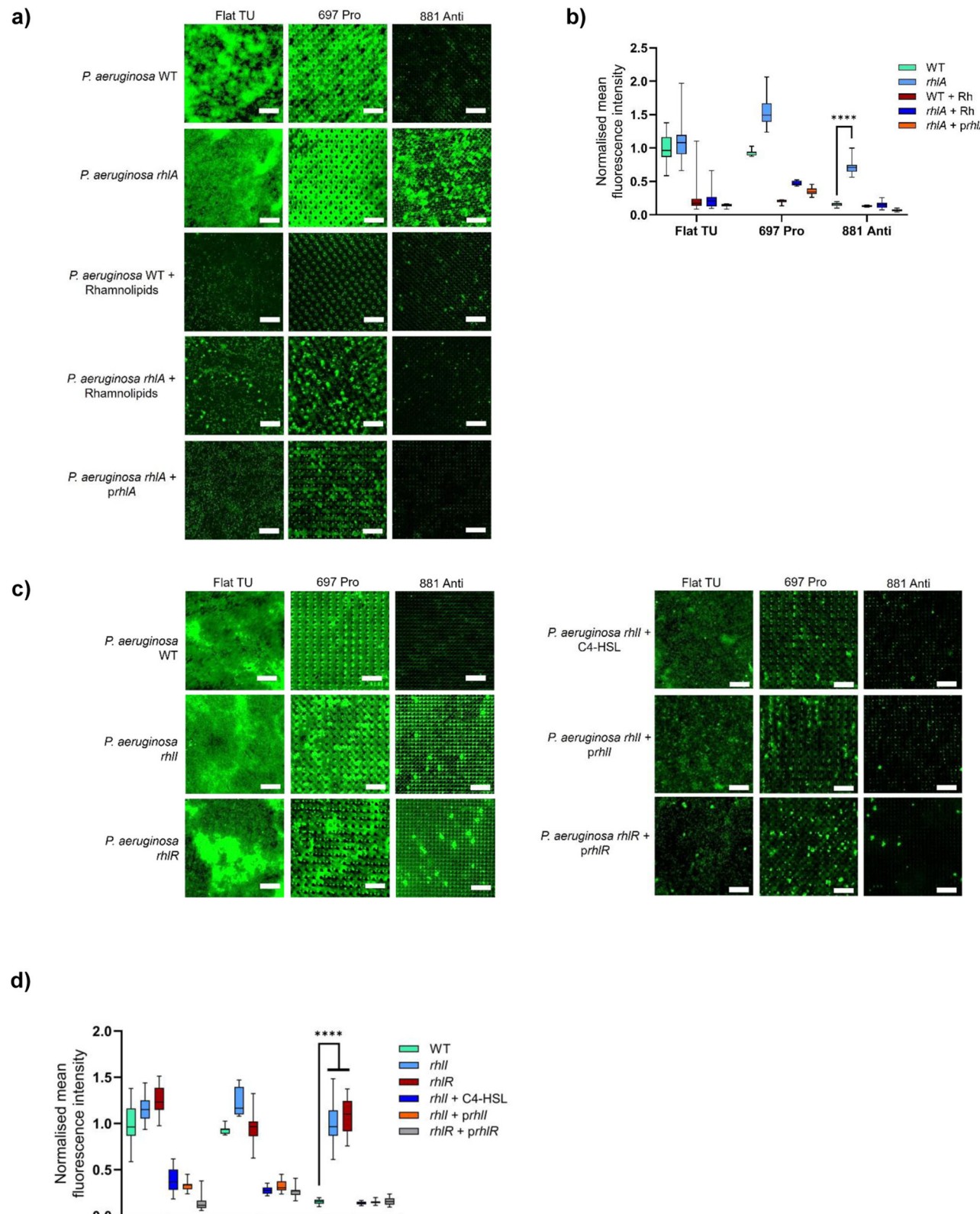

coefficients were associated with low levels of attachment. The most important descriptors were the inscribed circle radius (which relates to the space between the pillar features), the area of the topological features, and the average compactness of the pattern area. Moreover, similar biological performance was recorded for 'hit' micro-topographies on three different polymer materials incubated under different conditions, indicating that the changes in bacterial attachment were dependent on the topographical features rather than surface chemistry (at least for this limited set of materials). Remarkably, in vitro experiments involving a rod-shaped Gram-negative pathogen capable of flagella-mediated swimming motility (*P. aeruginosa*) and a Gram positive, round-shaped coccus (*S. aureus*) that is unable to swim

**Fig. 7 | Quorum sensing dependent rhamnolipid production prevents biofilm formation on anti-attachment TopoUnits.** Representative fluorescent images **a** and normalized mean fluorescence intensities **b** of *P. aeruginosa* wildtype and Δ*rhlA* without or with exogenous rhamnolipids and with Δ*rhlA* genetically complemented with *rhlA* on flat ($n = 20$ TopoUnits), pro ($n = 9$ TopoUnits) and anti-attachment ($n = 9$ TopoUnits) surfaces after 24 h incubation under static conditions. Data shown in boxes extend from the 25th to 75th percentiles and lines in boxes correspond to the median values. Whiskers go down to the smallest and up to the largest values. Statistical analysis was done using a two-way ANOVA with Dunnett's multiple comparisons test (*$p < 0.05$; **$p < 0.01$; ***$p < 0.001$; ****$p < 0.0001$). *P*-value WT vs *rhlA* on the 881 topography = 6.54E−13.

Representative images **c** and normalized mean fluorescence intensity **d** of *P. aeruginosa* wildtype, Δ*rhlI* without or with exogenous C4-HSL, Δ*rhlI* genetically complemented with *rhlI*, Δ*rhlR* and Δ*rhlR* genetically complemented with *rhlR* on flat ($n = 20$ TopoUnits), pro ($n = 4$ TopoUnits) and anti-attachment ($n = 16$ TopoUnits) surfaces after 24 h incubation under static conditions. Data shown in boxes extend from the 25th to 75th percentiles and lines in boxes correspond to the median values. Whiskers go down to the smallest and up to the largest values. Statistical analysis was done using a two-way ANOVA with Dunnett's multiple comparisons test (*$p < 0.05$; **$p < 0.01$; *** $p < 0.001$; **** $p < 0.0001$). P-value WT vs *rhlI* and *rhlR* on the 881 topography < 1.00E−15. Scale bar: 50 μm. For Fig. 7b, d, the source data are provided as a Source Data file.

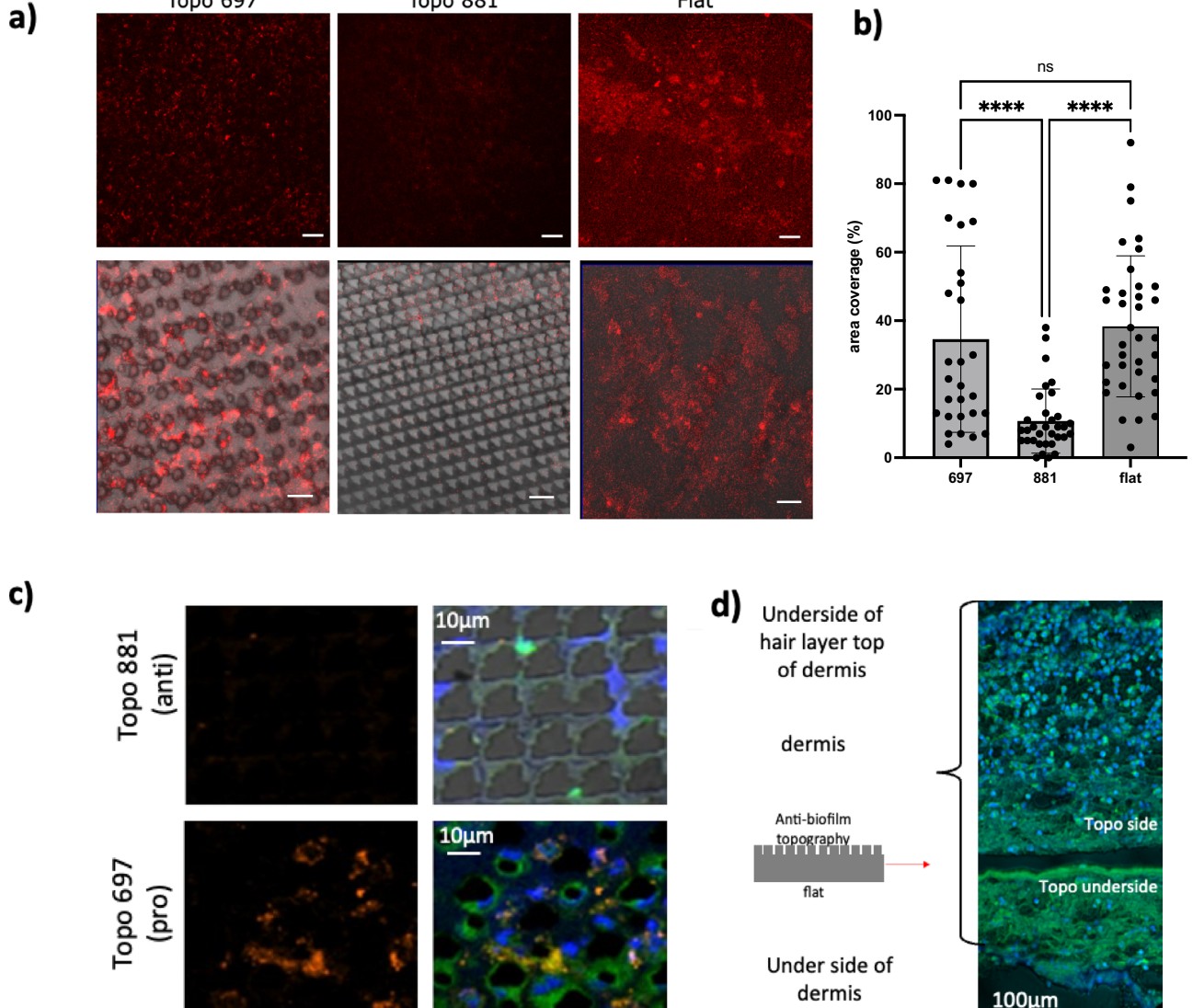

**Fig. 8 | Colonization of pro- (697) and anti- (881) attachment PU TopoUnits compared with flat PU implanted subcutaneously in mice.** After implantation, mice were allowed to recover for 4 days prior to infection with *P. aeruginosa* for a further 4 days. The implants were recovered and imaged via confocal fluorescence microscopy. **a** Representative images of *P. aeruginosa* PAO1 pcE2C (red) constitutively expressing the fluorescent protein E2Crimson. Upper panels, scale bar 50 μm; Lower panels, fluorescent images overlaid on brightfield images; scale bar, 20 μm. **b** Quantification of *P. aeruginosa* pcE2C surface coverage from individual explants (one per mouse) using constitutive E2C fluorescence. TopoUnits 881 ($n = 8$ mice) and 697 ($n = 7$ mice; one implant was unrecoverable) were compared with Flat PU ($n = 8$ mice). Statistical analysis was done using one way ANOVA with

Tukey's multiple comparison test. 697 vs 881, $p = <0.0001$ ****; flat vs 881, $p = <0.0001$ ****; flat vs 697 $p = 0.9328$, NS, not significant. The source data are provided as a Source Data file. **c** Representative images of explanted TopoUnits 881 and 697 stained for bacteria and host cells. Lefthand panels: orange, PAO1 bacteria detected using IHC antibody PA1-73116 and secondary anti-rabbit Alexa 555. Righthand panels: brightfield images of the lefthand panels stained in addition with DAPI (for DNA nuclei, blue) and FM1-43 (for cell membranes, green). Scale bars, 10 μm. **d** Sterile explant recovered after 4 days showing increased host cell migration on the upper side of TopoUnit 881 compared with the flat side indicative of an asymmetrical host response. Blue, DAPI, DNA stained nuclei (blue), FM1-43 stained membranes, green. Scale bar, 100 μm.

as it lacks a flagellum, showed that similar surface topographical descriptors were the most important. The responses of *A. baumannii* (non-flagellated) and *Pr. mirabilis* (flagellated) to selected pro- and anti-attachment TopoUnits were also very similar to those of *P. aeruginosa* and *S. aureus*. These findings illustrate the strength of unbiased screening of large topographical libraries to reveal previously unperceived cell–surface interactions as well as providing insights towards the rational fabrication of new bioactive surfaces[25,47,48].

Real time single cell tracking of *P. aeruginosa* on topographies exposed to growing cultures revealed that the bacteria could access all niches available within the micro-topographies selected. Within a given pattern, bacterial attachment depends on both the width and spacing of the topographical features and the size of the bacterial cell[22]. However, the feature sizes encountered by the bacteria on the TopoChip are significantly larger than bacterial cells (10, 20 or 28 μm). These observations suggested that the differences in attachment to different TopoUnits observed were not simply related to the bacteria being unable to access certain topographical areas but also to cells sensing and responding in different ways to the niches available within the different TopoUnits. Indeed, surface mechanical cues have previously been linked to variations in attachment-stimulated surface sensing driven by modulating c-di-GMP levels in *P. aeruginosa*, with both flagella and type IV pili involved in the process[31,32,49]. Our results revealed some minor differences in attachment to pro-attachment micro-topographies by *P. aeruginosa* type IV pilus (Δ*pilA*) and flagellum (Δ*fliC*) mutants, supporting a role for these appendages in exploration and adhesion to micro-topographically defined surfaces[46]. However, there were no obvious differences between the Δ*pilA*, Δ*fliC* mutants and the parent strain with respect to the anti-attachment TopoUnits.

In *P. aeruginosa*, the Wsp chemosensory surface sensing system, which is activated, at least in part, via surface-induced envelope stress[37] drives surface attachment and biofilm formation via elevated levels of the second messenger c-di-GMP. However, *P. aeruginosa wspF* mutants which constitutively synthesize elevated c-di-GMP levels have also been reported to overproduce rhamnolipids[50]. This may help explain the inability of the *wspF* mutant to form a biofilm on the anti-attachment topographies. Since the Wsp system is involved in the regulation of diverse metabolites as well as biofilm formation[50], we introduced the *E. coli* c-di-GMP synthase gene, *yedQ*, which has no other sensory domains and induces activation of biofilm development programmes in *P. aeruginosa* via c-di-GMP overproduction[38]. Expression of *yedQ* was also unable to facilitate biofilm formation on the anti-attachment TopoUnits indicating that their biofilm resistance was not associated with c-diGMP signalling pathways.

Within anti-attachment topographies such as 881, the trajectories of motile *P. aeruginosa* cells were typified by confinement within the 2 μm channels and accumulation at the feature crevices. Sub-micron crevices are known to impact on the attachment of *E. coli* where adhesion to topographical surfaces was reduced during the first 2 h but increased over longer periods coinciding with bacterially induced wetting transitions aided by attachment promoted via flagella filaments[46]. Our observation that *P. aeruginosa* cells swim within the channels between the 10 μm feature pillars of the anti-attachment TopoUnit 881 and become trapped by crevices in the features is consistent with rod-shaped cells having a greater propensity to become trapped at the bottom of high walled structures, a phenomenon that has been modelled for engineered steps[51]. The differences in bacterial localisation on anti- and pro-attachment topographical patterns highlights the importance of early-stage cell confinement for preventing biofilm development.

Rhamnolipid biosurfactants not only possess anti-adhesive and lubricating properties[40,41] but play important roles in the maintenance of *P. aeruginosa* biofilm architecture[52,53] and in the dispersal of mature biofilms[54]. Our data show that exogenous provision of purified

rhamnolipids prevented *P. aeruginosa* biofilm formation on pro-attachment TopoUnits and flat surfaces, the latter confirming similar observations by others[52]. On anti-attachment TopoUnits, abolition of rhamnolipid biosynthesis via deletion of the *rhlA* gene resulted in biofilm formation by *P. aeruginosa*. Since rhamnolipid production is regulated by QS at high bacterial population densities in liquid cultures, this suggested that the likely restricted diffusion of QS signal molecules in *P. aeruginosa* trapped in the 881 TopoUnit feature crevice microenvironment resulted in the induction of QS in a low population density leading to rhamnolipid production, so inhibiting early-stage biofilm formation. This is because the size of a 'quorum' is not fixed but depends on the relative rates of production and loss of the cognate QS signal molecule. In complex topographies containing cracks and crevices[45] or where bacteria become trapped in vesicles[55], QS signal molecules can accumulate and impact cell-cell communication. Consequently, *P. aeruginosa* cells trapped in the TopoUnit feature crevices can become quorate at lower population densities. Genetic evidence in support of this hypothesis was obtained by deletion of the genes coding for the QS signal synthase *rhlI* or the response regulator, *rhlR* or by provision of the QS signal molecule C4-HSL to the *rhlI* mutant which restored rhamnolipid biosynthesis. Furthermore, surface association is known to sensitize *P. aeruginosa* to QS[56] through induction of the QS master regulator LasR which controls the expression of both *rhlR* and *rhlI*[57].

While the *Acinetobacter*, *Proteus* and *S. aureus* strains investigated were also unable to form biofilms on anti-attachment micro-topographies, the molecular mechanism(s) involved are likely to differ from that of *P. aeruginosa*. Although these species do not produce rhamnolipids, they possess QS systems[55,58,59] that may contribute to their inability to form biofilms on the anti-attachment TopoUnits.

To determine whether the pro- and anti-attachment properties of the TopoUnits were maintained in complex environments where the TopoUnit surfaces are likely to be conditioned by host proteins and cells, we examined their response to *P. aeruginosa* after conditioning with human serum in vitro or after subcutaneous implantation into mice in vivo. In both cases, the anti-attachment properties were maintained, suggesting that the deposition of host proteins and cells did not alter the biofilm resistant properties of the anti-attachment TopoUnits. Consequently, these micro-topographies have considerable potential for preventing biofilm formation in a clinical context. Texturizing the surface of clinically approved biomaterials would have the benefit of retaining their physical and mechanical properties while significantly reducing the need for expensive new materials discovery and commercial development. In addition, since material topography is known to influence macrophage attachment and phenotype[19] and our in vivo experiments support the possibility of a productive host response, this work raises the exciting prospect of exploiting micro-topographies to modulate host immune responses and prevent both biofilm-centred infections and foreign body rejection of implanted medical devices, while greatly reducing the occurrence of resistance observed for antimicrobial-eluting biomedical materials.

## Methods
### Ethics statement
This research complied with all relevant ethical regulations. All animal studies were approved following local ethical review at the University of Nottingham and performed under U.K. Home Office Project Licence 30/3082.

### Bacterial strains and culture conditions
Bacterial pathogens commonly associated with medical device-associated infections[60] chosen for this study included the Gram-negative species, *P. aeruginosa*, *Pr. mirabilis* and *A. baumannii* and the Gram-positive, *S. aureus*. The strains used are listed in Supplementary Table S4. Bacteria were routinely grown at 37 °C in lysogeny broth (LB)

or LB agar supplemented with antibiotics as required. In order to maintain the same culture conditions for each bacterial species, tryptic soy broth (TSB) was used as the growth medium for all bacterial TopoChip assays. To mimic in vivo conditions, TSB supplemented with 10% v/v human serum (TSBHS10%) was used for some experiments. Where required, purified rhamnolipids (100 ug/mL, R90-10G Sigma Aldrich) or C4-HSL (20μM; synthesized in house[61]) were added to the growth medium. For PU TopoChip attachment assays, *P. aeruginosa* PAO1 carrying the constitutively expressed *mCherry* gene on the plasmid pMMR (Supplementary Table S4) was used to avoid PU autofluorescence.

## Construction of *P. aeruginosa* PAO1 deletion mutants

*P. aeruginosa* PAO1 ΔpilA, ΔfliC, ΔrhlA, ΔrhlI and ΔrhlR mutants were constructed via 2-step allelic exchange obtained by double crossover using pME3087 or pEX18Gm^R vectors. Two PCR products amplifying the upstream and the downstream nucleotide regions of each gene were generated using the primer pair 1FW/1RW and 2FW/2RW, respectively (Supplementary Table S4). For each mutant, the PCR products were fused by overlapping PCR to create a deletion in the corresponding gene and the resulting fragment was cloned into the suicide plasmid pME3087 (Supplementary Table S4). Integration of the suicide plasmid into the *P. aeruginosa* chromosome was carried out by conjugation. Recombinants were selected on tetracycline (125 μg ml⁻¹). The second cross over event was carried out using the carbenicillin enrichment method (300 μg ml⁻¹) to select for the loss of tetracycline resistance[62]. The resulting colonies were screened for the loss of antibiotic resistance by plating on LB supplemented with or without tetracycline. To construct the *P. aeruginosa* PAO1 ΔrhlA mutant, the two PCR products amplifying *rhlA* upstream and downstream nucleotide regions were ligated together in pBluescriptII KS+ (VWR) to create the *rhlA* deletion and subsequently cloned into the suicide vector pEX18Gm^R (Supplementary Table S4). The resulting *P. aeruginosa* colonies were screened for the loss of antibiotic resistance by plating on LB supplemented with or without tetracycline or gentamicin. The in-frame deletions were confirmed by PCR and DNA sequence analysis and the rhamnolipid phenotype for each mutant compared with wild type was quantified using the orcinol assay[63]. The loss of C4-HSL production by the ΔrhlI mutant was confirmed by LC-MS/MS[61] (Supplementary Fig. S8b).

## Construction of genetic complementation vectors

Genetic complementation of *P. aeruginosa* ΔpilA, ΔfliC, ΔrhlA, ΔrhlI and ΔrhlR, was carried out using the expression vector pME6032ΔlacIQ. The relevant gene was amplified by PCR using the Complem F/Complem R primer pair (Supplementary Table S4). The fragments were cloned into pME6032ΔlacIQ and introduced into *P. aeruginosa* mutant strains by electroporation. For the ΔpilA and ΔfliC mutants, genetic complementation was confirmed by swimming and twitching assays (Supplementary Fig. S8c). The restoration of rhamnolipid production for each of the complemented strains was confirmed as was C4-HSL production by the ΔrhlI complemented mutant (Supplementary Fig. S8a).

## TopoChip microtopography library

The TopoChip was designed by randomly selecting 2176 features from a large in silico library of features containing single or multiple 10 μm high pillars within a virtual square of either 10 by 10, 20 by 20, or 28 by 28 μm² size[12]. Micro-pillars were constructed from three different microscale primitive shapes: circles, triangles and rectangles (3–10 μm widths). Topographies were assembled as periodical repetitions of the features within 290 × 290 μm micro-wells (TopoUnits) surrounded by 40 μm tall walls in a 66 × 66 array containing duplicate TopoUnits for each topography together with flat control surfaces. Each micro topography was assigned a feature index (FeatIdx) number (from 1 to

2176). TopoChips were fabricated on a 2 × 2 cm² chip divided into four quadrants as previously described[12] (Supplementary Figs. S1 and S2a). Briefly, the inverse structure of the topographies was produced in silicon by standard photolithography and deep reactive etching. A mould fluorosilane release agent, trichloro (1H, 1H, 2H, 2H-perfluorooctyl) silane (Sigma Aldrich) was applied using vapour deposition. This silicon mould was then used to make a positive mould in poly(dimethylsiloxane) (PDMS) from which a second negative mould in OrmoStamp hybrid polymer (micro resist technology Gmbh) was fabricated. This served as the mould for hot embossing PS (DSM), PU (Elastollan, GoodFellow) and cyclic olefin copolymer films (Good-Fellow) to produce TopoChips with 3 different chemistries. The same hot embossing methodology was used to produce PS and PU TopoUnit scale-ups incorporating 4 replicates of 8 micro-topographical patterns for cell tracking and mouse infection experiments. After fabrication TopoChips and scale-ups were subjected to oxygen plasma etching for 30 s.

## Surface chemical analysis

To ensure that the bacteria-material interactions observed were specifically dependent on surface topography, TopoChips were analysed by time-of-flight secondary ion mass spectrometry (ToF-SIMS) together with X-ray photoelectron spectroscopy (XPS) for quantitative elemental analysis. ToF-SIMS measurements were conducted on an OrbiSIMS instrument (HybridSIMS, IONTOF GmbH) instrument using a monoisotopic Bi₃⁺ primary ion source operated at 30 kV with a time-of-flight analyser applying 'delayed extraction mode'. Charge compensation was performed using a low energy (20 eV) electron floodgun and data acquisition and analysis was conducted using SurfaceLab7 (IONTOF GmbH). Analysis was conducted over areas of 50 × μm and 150 x 150 μm acquired at a resolution of 256 × 256 pixels by rastering the primary ion beam over the patch using a 'random raster' path sequence. TopoChip surface chemistry was quantified in terms of elemental composition using an Axis-Ultra XPS instrument (Kratos Analytical, UK) with a monochromated Al kα X-ray source (1486.6 eV) operated at 10 mA emission current and 12 kV anode potential (120 W). Small spot aperture mode was used in magnetic lens mode (FoV2) to measure a sample area of approximately 110 μm². CasaXPS (version 2.3.18dev1.0x) software was used for quantification and spectral modelling. The measured N 1s fraction for growth medium serum-conditioned surfaces was converted into protein layer thickness using Ray and Shard[64] relationship between [N] and protein depth. A low level of mould release agent (F = 2 at%) was found to be transferred to the surface by the TopoChip production process (Supplementary Fig. S3a, b) which was unavoidable due to the moulding requirements. However, it was clear from the chemical images that this was distributed evenly in the TopoChips and therefore were deemed suitable for investigating specific bacterial responses to the microtopographies.

## TopoChip screen for bacterial attachment and biofilm formation

Prior to incubation with bacteria, TopoChips were washed by dipping in distilled water and sterilized in 70% v/v ethanol. The air-dried chips were placed in petri dishes (60 mm × 13 mm) and incubated statically at 37 °C in 10 ml of growth medium inoculated with actively growing bacteria for 4 h or 24 h (from optical density: OD₆₀₀ ₙₘ = 0.01). It should be noted that despite air bubbles forming on the arrays surface after immersion in growth medium, air pockets were removed by brief incubation at 37 °C and repeated media pipetting until no entrapped bubbles between the topographical features could be detected. In addition, for narrowly spaced TopoUnit patterns such as 881, individual bacterial cells were clearly observed swimming along the 2 μm channels between the pillars (See Fig. 5 and Supplementary Movie 1). Static incubation was selected for initial screening experiments as this

produced consistent bacterial attachment while reducing the formation of biofilm streamers on TopoChip corners which can induce cross contamination of TopoUnits under flow[65]. To explore the impact of gravity and flow, for some experiments, the TopoChips were inverted or shaken (at 60 rpm in an orbital shaker). At specific time points, TopoChips were removed and washed in phosphate-buffered saline (PBS) pH 7.4 to remove loosely attached cells. After rinsing with distilled water, attached cells were stained with Syto9 (50 μM; Molecular Probes, Life Technologies) for 30 min at room temperature. After staining, TopoChips were rinsed with distilled water, air-dried and mounted on a glass slide using Prolong antifade reagent (Life Technologies). All experiments were performed at least 3 times unless otherwise indicated.

### TopoChip imaging and data acquisition
TopoChips were imaged using a Zeiss Axio Observer Z1 microscope (Carl Zeiss, Germany) equipped with a Hamamatsu Flash 4.0 CMOS camera and a motorized stage for automated acquisition. A total of 4356 images (one per TopoUnit) were acquired for each chip using a 488 nm laser as light source. Since the bacterial cells may attach at different heights on the micro-patterns, images were initially acquired as 50 μm range Z-stacks (2 μm steps - 25 slides) from the TopoUnits using a 40× objective (Zeiss, LD Plan-Neofluar 40×/0.6 Korr Ph 2 M27). Although this method facilitates capture of the total fluorescence emitted by bacteria attached to the topographies, it significantly increased scanning times and file sizes. Hence, a lower ×10 magnification lens (Zeiss, EC Plan-Neofluar 10×/0.30 Ph 1) was routinely used as it provided sufficient depth resolution for capturing the total fluorescence per TopoUnit. This enabled the use of the auto-focus function, which considerably reduced scanning times and file sizes per TopoChip. Cropping the images into a smaller field of view (247 μm × 247 μm) and omitting the walls of the micro-wells reduced the artefacts arising from bacterial attachment to the walls and further improved the auto-focus function.

To identify out-of-focus images from each TopoChip dataset, individual topographical images were combined into composites using open source Fiji-ImageJ 1.52p software (National Institutes of Health, US). To improve TopoChip dataset quality, image pre-processing also included a) staining artefact removal by excluding pixels with fluorescence intensities higher than 63,000 from data acquisition and b) assay-specific background removal by applying the automatic thresholding function of Fiji-ImageJ to flat control images and the average minimum threshold value obtained used as baseline to segment all images from the same experiment.

To classify topographies influencing bacterial attachment, the mean fluorescence intensity on each TopoUnit was measured using Fiji-ImageJ. Hit micro-topographies with anti-attachment properties were selected for further studies based on the screening data obtained from quantifying *P. aeruginosa* ($n = 22$) and *S. aureus* ($n = 6$) attachment to PS TopoChips (Supplementary Fig. S1a). Additionally, and to discriminate between bacterial responses to hit topographies with other patterned surfaces, TopoUnits with similar or small increases in attachment with respect to flat controls ("pro-attachment TopoUnits") were also selected for further investigation. To ascertain that hit surfaces influenced the observed bacterial responses, two-way ANOVA with Dunnett's multiple comparisons tests were applied to determine whether bacterial attachment on TopoUnits differed significantly from that of a flat control ($p < 0.05$) when compared with the variations within the replicates using GraphPad Prism 8.0 (GraphPad Software, Inc., San Diego, CA).

### Generation of predictive surface topography attachment models
To define the surface design parameters that influence bacterial attachment, the data acquired from screening *P. aeruginosa* and *S.*

*aureus* on PS TopoChips were interrogated as follows. The fluorescence values for bacterial attachment to each of the replicate TopoUnits were normalised to the average fluorescence intensity of the chip to account for differences in staining intensities between experiments. The fluorescence intensity was established to correlate with the number of attached fluorescent bacteria, as shown previously[15]. It was therefore used as the dependent variable in the models. TopoUnits with low signal to noise ratio (<2) were excluded from the datasets of *P. aeruginosa* (77 units removed) and *S. aureus* attachment (10 units removed). The Random Forest machine learning method was applied to generate relationships between the topographies and bacterial attachment using the CellProfiler topographical descriptors listed in Supplementary Table S2. The Random Forest module was used with default parameters in Python 3.7. Bootstrapping without replacement was used to define training and test sets for 50 model runs. Seventy percent of each bacterial attachment dataset was used to train the model, and 30% were kept aside to determine the predictive power of the model. The SHAP (SHapley Additive exPlanations) package in Python 3.7 was used to eliminate less informative descriptors and to determine descriptor importance.

### Bacterial single-cell tracking
To track single bacterial cells interacting in 3D with flat, pro- and anti-attachment TopoUnits, a bespoke Nikon Eclipse Ti inverted multi-mode microscope (Cairn Ltd.) equipped with a Hamamatsu Orca Flash 4.0v2 sCMOS camera and an OKOLab cage incubation system to maintain temperature (37 °C) and humidity constant (95%) was used. Differential interference contrast (DIC) imaging was carried out using a single channel white MonoLED (Cairns) light source and a 40× objective (Nikon, CFI Plan Fluor 40×/1.3). High speed movies were made up of images acquired for 20 ms for 1000 frames (total image sequence of 20 s). Experiments were conducted using a custom designed single well holder (22 mm × 22 mm × 50 mm depth) in static growth conditions[66]. PS topographies were adhered to the holder using an adhesive imaging spacer (Grace Bio Labs 654008). Surfaces were preconditioned with 4 mL of TSBHS10% medium for 30 min prior inoculation with mCherry-labelled *P. aeruginosa* (OD$_{600nm}$ 0.02). Imaging began within ~2 min of adding the bacterial suspension.

Image analysis was undertaken using Fiji-ImageJ. Sequences of images were registered to account for any drift using Descriptor-based Series Registration[67]. The series of images were divided by the median projection of the image series to remove the topographies (Supplementary Fig. S1c). This procedure was essential to allow individual cells to be observed and tracked in narrow channels, since the topographies and their light distortion obscures the cells. Trainable Weka Segmentation v3.3.2[68] was used to segment images and produce a probability map of bacteria locations across the image series. A maximum entropy[69] threshold was applied to the probability map series, and this was then used for tracking. Tracking was done in TrackMate v7.9.2[70] and bacterial tracks could be categorised as swimming or stationary. Swimmers were defined by a maximum distance travelled (>3 μm), displacement (>4 μm) and mean directional change (<90 AU). All experiments were performed at least 3 times unless otherwise indicated.

### Murine foreign body infection model
To investigate the progress of bacterial infection and the host immune response to pro- (FeatIdx 697) and anti- (FeatIdx 881) attachment TopoUnits, a murine foreign body infection model was used[15]. Scaled up TopoUnit micro-patterns were embossed on one side of a PU sheet of dimensions 7 mm × 2 mm. TopoUnits or flat PU with the same dimensions were implanted subcutaneously using a 9-gauge trocar needle (1 per mouse, $n = 8$ for each topography; sample size was based on those described in Hook et al.[15]) into 19–22 g female BALB/c mice (Charles River Laboratories) with the patterned side facing upwards to

the skin surface. One hour before implantation, 2.5 mg/kg of Rymadil analgesic (Pfizer) was administered by subcutaneous injection. Animals were anaesthetized using isoflurane, the hair on one flank removed by shaving and the area sterilized with Hydrex Clear (Ecolab). After foreign body insertion, mice were allowed to recover for 4 days prior to subcutaneous injection of either $1 \times 10^5$ colony forming units (CFUs) of fluorescent *P. aeruginosa* pcE2C (Supplementary Table S4) or vehicle (phosphate-buffered saline; uninfected control).

For the in vivo experiments, mice were randomized into groupings of 4 per cage; investigators were not blinded to allocation during experiments and outcome assessment. Mice were housed in individually ventilated cages under a 12 h light cycle, with food and water *ad libitum*, and with weight and clinical condition of the animals recorded daily. Four days post infection, the mice were humanely killed and the micropatterned PU TopoUnit and flat samples and the surrounding tissues removed. Bacterial coverage on the explants was quantified via confocal fluorescence microscopy with 4 regions of interest captured per explant. Images were analysed using Fiji-ImageJ, by initially subtracting the background, z-projection to maximum intensity followed by manual thresholding to capture the bacterial coverage (%). Statistical analysis was done using one way ANOVA with Tukey's multiple comparison test PU explants and tissues were also fixed in 10% v/v formal saline and labelled with the DNA stain DAPI and the membrane-selective dye, FM1-43 (Thermofisher Scientific). Bacterial cells on the fixed explant surfaces were visualized with polyclonal antibodies raised against *P. aeruginosa* (Thermofisher Scientific) and detected using anti-rabbit Alexa 555 (Thermofisher) as secondary antibodies.

### Reporting summary

Further information on research design is available in the Nature Portfolio Reporting Summary linked to this article.

## Data availability

Data supporting the findings of this study are provided within the manuscript and its associated Supplementary Information/Source Data file. Source data are provided with this paper.

## Code availability

For the machine learning models, the code is available at https://github.com/Biomaterials-for-Medical-Devices-AI/Helix.

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

## Acknowledgements

This work was supported by the Engineering and Physical Sciences Research Council [grant nos. EP/N006615/1, EP/X001156/1, EP/P029868/1 and EP/K005138/1] the Wellcome Trust [grant nos. 103882 and 103884], the Biotechnology and Biological Sciences Research Council [BB/R012415/1], the European Union Horizon 2020 Programme (H2020-MSCA-ITN-2015; Grant agreement 676338], the Dutch Science Foundation (NWO) [grant VENI 15075], and the Dutch province of Limburg. M.R. was also supported by the Maria Zambrano program and

Research Consolidation grant (CNS2023-145299) from the Spanish Ministry of Science, Innovation and Universities. We thank Dr Emily F Smith at the Nanoscale and Microscale Research Centre (NMRC - University of Nottingham) for acquiring XPS spectra and Dr Marta M Paino with XPS data interpretation. TopoChips were imaged in the School of Life Sciences Imaging Unit (SLIM - University of Nottingham). We thank Nick Beijer and Nadia Roumans for TopoChip fabrication assistance, Chris Gell and Arsalan Latif for their help with data acquisition.

## Author contributions

M.R.A., J.deB. and P.W. conceived the project. M.R., J.F.D., E.I., A.M.C. and A.H. conducted the in vitro, and J.L. the in vivo experiments. J.F.D. constructed the *P. aeruginosa* mutants. J.L., L.K., A.daS. and A.M.G. analysed the in vivo data. G.P.F., A.uC., A.V. and D.A.W. conducted the modelling and the machine learning. S.V., P.K.S., X.X. and C.B. fabricated the TopoChip arrays and scale-ups, D.J.S. analysed the TopoUnit surface chemistry. P.W., M.R. and M.R.A. wrote the manuscript with input from all the other authors.

## Competing interests

The authors declare no competing interests.

## Additional information

Manuel Romero [1,2,8], Jeni Luckett[1], Jean-Frédéric Dubern [1,2], Grazziela P. Figueredo[3], Elizabeth Ison[1,4], Alessandro M. Carabelli [4], David J. Scurr [4], Andrew L. Hook [4], Lisa Kammerling[1], Ana C. da Silva[1,2,9], Xuan Xue[4,10], Chester Blackburn[4], Aurélie Carlier [5], Aliaksei Vasilevich[6], Phani K. Sudarsanam[6], Steven Vermeulen [5,6], David A. Winkler [7], Amir M. Ghaemmaghami[1], Jan de Boer[6], Morgan R. Alexander [4] ✉ & Paul Williams [1,2] ✉

[1]School of Life Sciences, University of Nottingham, Nottingham, United Kingdom. [2]National Biofilm Innovation Centre, Biodiscovery Institute, University of Nottingham, Nottingham, UK. [3]School of Computer Science, University of Nottingham, Nottingham, United Kingdom. [4]School of Pharmacy, University of Nottingham, Nottingham, United Kingdom. [5]MERLN Institute for Technology-Inspired Regenerative Medicine, Maastricht University, Maastricht, The Netherlands. [6]Department of Biomedical Engineering, Eindhoven University of Technology, Eindhoven, The Netherlands. [7]Department of Biochemistry and Genetics, La Trobe Institute for Molecular Science, La Trobe University, Melbourne, VIC, Australia. [8]Present address: Department of Microbiology, Faculty of Biology-Aquatic One Health Research Center (iARCUS), Universidade de Santiago de Compostela, Santiago de Compostela, Spain. [9]Present address: Cancer Research UK Cambridge Institute, University of Cambridge, Li Ka Shing Centre, Cambridge, United Kingdom. [10]Present address: Department of Chemistry, School of Science, Xi'an Jiaotong - Liverpool University, Suzhou, PR China. ✉e-mail: morgan.alexander@nottingham.ac.uk; paul.williams@nottingham.ac.uk

