## [Transparent Peer Review file · Nature Communications]

Combinatorial discovery of microtopographical landscapes that resist biofilm formation through quorum sensing mediated autolubrication

Corresponding Author: Professor Paul Williams

Version 0:

Reviewer comments:

Reviewer #1

(Remarks to the Author)

In this study, the authors screened a library of μm size patterns and used machine learning to generate design rules for anti-biofilm surfaces. The work addresses an important problem and it is plausible to use ML in this area. However, the results contradict the general observations by other labs and did not investigate the mechanism. Thus this reviewer felt it is premature to publish in Nature Communications.

Specific comments:

The results show that less cells attach to the surfaces with narrowly spaced patterns. As the authors acknowledged, this is the opposite of other studies. In addition to refs. 18 and 19 mentioned, Ref 30 (Friedlander et al.) cited in this paper showed how bacterial flagella interact with topographical features and the role of pattern spacing. The results showed closely positioned pillars promote biofilm formation, unlike reported here. While it is good to present different results, the authors did not offer an explanation or speculation about the possible mechanism/cause of the differences. For example, is that possible that some air was trapped between the close features and thus led to the Wenzel state and inhibited bacterial attachment? If not, was there any treatment of the samples to prevent that?

The in vivo study gave 4 days between implantation and bacterial inoculation. This allows the host cells to cover the surface and thus it is not clear if the inoculated bacteria can still have access to the patterns. A shorter gap can help better see the effects.

Some literature citations are incorrect. For example, Ref. 30 has μm size patterns, not nanopatterns as the authors claimed. There are also other studies on topographical features not cited here. It is impractical to cite all, but giving a better covered Introduction can help the audience understand the field.

It is surprising that, although different shapes were tested, only the descriptors of circles are among the top 5 identified. Some explanation/speculation would help.

P7, l8. The authors claimed that their new patterns are better than Sharklet. This direct comparison is inappropriate since the conditions are different. The cited test of Sharklet was 4 days in rich media and 24 h in minimal media, while this study tested 4 hour attachment in a rich medium. In addition, the ref did report Sharklet biofilm control for both MASA and P. aeruginosa. The statement that Pa results of Sharklet are not available is incorrect.

The two most effective designs (881 and 685) are only shown in a small image in Figure 4. The actual dimensions and possible mechanism of biofilm reduction should be described. It is also important to describe the detailed results of other similar patterns (same size but different shape/spacing or same spacing but different size/shape).

The video links do not connect to the specific video files but rather the website of the team.

P11, 24-26. This statement is not true. For example, <https://www.nature.com/articles/s41551-021-00739-4>.

The resolution of Fig. 2 is low.

L11, l10-12. It is unclear how this statement can be made since the surface is already covered by host proteins and cells.

Reviewer #2

(Remarks to the Author)

The paper proposed polymeric surface patterns that dramatically limit bacterial pathogens adhesion. By using a high-throughput unbiased screen of micro-topographical patterns on polystyrene derived from the assembly of primitive designs, authors identified specific patterns that significantly prevent the adhesion of 2 important bacterial pathogens (*Pseudomonas aeruginosa* and *Staphylococcus aureus*). Despite important differences between these two species (motility, cell morphology, cell wall decorations...) the correlation between the behaviour of these two species suggests the robustness of the antiadhesive effect. This robustness was extended with different materials, two other bacterial species, in an in vitro biofilm assay and finally in a murine model. A machine learning approach allows pinpointing specific traits of antiadhesive patterns that could be integrated into the development of bioactive surfaces.

The general strategy and the efficacy of the micro-patterned antiadhesive surfaces are impressive. The paper is well constructed, and well written and the presented data are in accordance with the conclusions.

I have however a few concerns with the article:

-in some figures, while the fluorescence images presented suggest almost no bacterial cells on the antiadhesive micro-patterned surfaces compared to the flat control, the quantification of the reduction appeared less contrasted. For example, on page 8 line 13 it is mentioned a 5-fold average reduction for *S. aureus* and in figure 4C the images presented suggest a higher level of reduction (the biofilm is almost covering the control and no cells are visible on the anti-adhesives surface). Is it associated with the selection of the presented images? Or to variation of fluorescence intensity on the different materials? In that case, could a segmentation step before the quantification (determination of the biovolume) be relevant?

- Page 8 line 29 and figure S5: what dyes were used for the live/dead staining, syto9 and propidium iodide (PI) proposed in the live/dead labelling? The mean fluorescence intensity presented in the control surface is very similar for the "live" and "dead" populations. How were these populations calculated: while PI is labelling only the dead cell with compromised integrity of their membranes, Syto 9 is labelling both live and dead populations. Quantification of live and dead populations is not directly the fluorescence signal of syto 9 and IP. Authors should clarify, either by explaining how they extracted the live and dead proportions or by presenting the raw signals with Syto9 and IP (and not extrapolating the relation with live and dead cells in the figure).

Then, if 50% of the initial adherent population is dead on the controls as suggested here, the authors should discuss this unexpected observation.

-The polymeric chip design and the general strategy are very similar to reference 17 published in PNAS in 2011. The main difference is the target cells (eucaryotic cells in the PNAS paper, bacteria here). While the results obtained with bacteria are impressive, this lowers a bit the novelty of the approach to generate those patterns.

-Page 4 line 27: "... early cell attachment could be prevented and AND HENCE BIOFILM FORMATION after longer incubation times". Please modulate this sentence: there is no universal correlation between the number of adherent cells and the biomass of the mature biofilm.

-Minor comments:

-On page 4 line 13, remove the symbol between "*Pseudomonas*" and "*aeruginosa*"

-On page 11 line 17, remove the symbol between (top) and "and flat"

-page 20 figure 4D: the choice of the scale of the 2 Y-axis appears arbitrary; could the authors find a way (e.g. normalize the fluorescence on the control flat surface) to compare the 2 strains?

-Page 22 Figure 6a: Here also, the images presented are not in concordance with the quantitative data presented in Figure 6b (the pro surface 697 appears far less contaminated than the flat control on the fluorescent and not on the average histogram). images while the quantification.

Reviewer #3

(Remarks to the Author)

Overall, the idea of screening topographical designs computationally is very interesting, but limited to known descriptors. Hence, I am not convinced that bioinspired materials, as presented here, have a real-life impact to reduce the burden of medical device-associated infections (as this was the focus that the authors have chosen). The materials reduced adherence but not biofilms entirely and their ideas seem to be preliminary at this point as long term studies are missing – it will be important to see whether the material is indeed of practical use. After reading this manuscript, I think the real-world application of improved materials is rather in the industry (e.g., biofouling) than in the health sector, as it could reduce cleaning costs / frequencies etc.

A 15-fold reduced colonization is not striking (although it might actually be higher depending on how to interpret the data). The mentioning of motile and non-motile bacteria should be rephrased, as there is enough evidence that *S. aureus* performs certain motility behaviours and *P. aeruginosa* would prefer to attach to an air-liquid interface (which was not studied) rather than submerged material (where gravity would pull them down anyways). Also, *P. aeruginosa* is usually not a good biofilm-

former in nutrient rich broth such as TSB (as bacteria naturally detach after a certain amount of time); which should be considered when interpreting the overall significance of the data. Moreover, the experimental setup for biofilm attachment is somewhat problematic as well, as diluted overnight cultures were used. For example, if *P. aeruginosa* overnight cultures were in late stationary phase, there is a very high chance that they won't express flagella (which would be required for immediate attachment in the new growth conditions). In this regard, the rationale of flagella and pilus mutant experiments are rather weak as they might not even be present in the wild type. In addition, one would at least expect to have complementation studies as well. But, I would have appreciated if they authors would have shown a biofilm overproducer strain rather than these ko-mutants. The other missing link would be experiments with increasing cell numbers; what happens if more bacteria are used – does this affect attachment? I really would have appreciated bacterial counts to confirm the presence (or absence) of bacteria. The quantitative representation is not providing the full picture.

The in vivo experiments are somewhat confusing as well. Figure 6b shows individual explants – does this mean individual mice? What was the power calculation for these experiments? It almost appears that data was 'forced' to be 'significant' as more than twice the amount of explants were used for the control vs Topunits. Since the *P. aeruginosa* strain PAO1 was used in the in vivo studies I was wondering about the mortality rate and how often did the mice cleared the infection?

Minor:

Figure 4 does not show studies with Sa as indicated in the legend and is missing complementation studies. The annotations in the figure are also wrong as the delta sign is missing

Version 1:

Reviewer comments:

Reviewer #1

(Remarks to the Author)

The authors conducted additional experiments and significantly revised the manuscript. I have a few additional comments:

Line 346. Is this decrease statistically significant? If so, the bar graph should show the * signs of related comparisons and the strains of genetic complementation should also be tested.

Line 359. Has the timing of c-di-GMP production in the WT strain been considered? To compare with the results of mutants, it is important to know the level of c-di-GMP in the WT under the experimental condition of this study.

There is some extensive discussion about the possible role of QS in the observed effects on attachment. It would be helpful to test a QS reporter strain to see if the trapped cells indeed have a higher level of QS activities.

Reviewer #2

(Remarks to the Author)

The authors have adequately addressed my comments, and the article is now in an appropriate form for publication.

Reviewer #4

(Remarks to the Author)

These reviewers were asked to focus on the rigor of the machine learning aspects of the submitted paper.

Providing some overarching feedback that relates to the paper in general and impacts upon the machine learning aspects to begin- whilst the paper seeks to provide a new approach to enhancing understanding of bacterial growth (and prevention of) on medical devices and materials substrates, a number of areas are unclear and require addressing. The paper states that "Attempts to prevent such infections have predominantly focused on the incorporation of leachable bactericidal antimicrobials into medical devices"; this is not strictly the case, with significant work also being done into other methodologies with both bacteria and cells. This large body of work is also at odds with the statement "Mechanistic underpinnings to cellular responses to materials and their chemical properties are lacking, preventing reliable de novo prediction and design of improved materials". The use of cells, bacterial cells, and similar terminology adds to this unclear message, as the reader can be confused as to what specific previous research and known information is being referred to. With the above in mind, a stronger, clearer rationale for the use of machine learning would benefit the paper including what specific gaps in knowledge and research are being addressed, and how, as well as how this will translate into the field.

Focusing on the Machine Learning studies, the research is attempting to predict the degree of bacterial attachment to specific surface topographies on polystyrene TopoChips. By analyzing various topographical features (such as feature coverage, gap size between topographies, and maximum feature radius), the study aims to identify which design parameters influence bacterial attachment. The goal is to establish "design rules" that can be applied to create surface structures that reduce bacterial attachment, potentially aiding in the development of materials resistant to biofilm formation by bacterial species like *Pseudomonas aeruginosa* and *Staphylococcus aureus*. The fluorescence intensity measured on each TopoUnit

serves as a proxy for the quantity of bacteria attached to the surface. Machine learning models (primarily Random Forest) are used to correlate topographical descriptors with bacterial attachment levels, with the prediction outcome being the fluorescence intensity that indicates bacterial attachment for each topographical design tested.

Although the idea is very interesting, we think that there many aspects that need to be clarify, justify and addressed before being considered to be published.

1. Lack of Methodological Justification: There is not any justification why Random Forest or MLREM were used, other models like gradient boosting (e.g., XGBoost), SVMs, or neural networks could be more suited to the dataset's structure and the prediction task.

Moreover, the choice of Random Forest default parameters is stated but not justified. This omission limits reproducibility because default parameters may yield different results in other implementations or environments. Moreover, without information on parameter tuning or optimization, it's challenging to determine if these default settings are optimal for this dataset.

2. Statistical analysiss: The text describes reducing 242 shape descriptors to 68, but the rationale for setting a Pearson correlation threshold of 0.85 isn't clearly justified. Including a justification for this specific threshold or comparing it to different thresholds could enhance the understanding of descriptor selection. Moreover, all the statistical analysis done should be explained specifically in the methods in a section called "Data Analysis and Preprocessing" and all the results shown or in the manuscript or in the supplementary material.

Additionally, from the manuscript provided the study did not perform an explicit exploratory analysis, normality test, or homoscedasticity test on the data. These steps are crucial and would enhance the study in several ways:

a) Exploratory Data Analysis (EDA)

Feature Distributions: EDA would allow researchers to understand the distribution of each topographical descriptor and fluorescence intensity variable. This could reveal patterns like skewness, outliers, or distinct clusters, which may affect model accuracy and interpretability. If skewed data is identified through EDA, transformations (e.g., log or square root) could be applied to stabilize variance and improve model predictions.

Descriptor Relationships and Correlations: EDA could visualize correlations between descriptors, clarifying whether the cut-off threshold for removing highly correlated features (Pearson correlation > 0.85) is suitable. For instance, a lower or higher threshold may capture more relevant descriptors or remove noise, potentially increasing model robustness. Additionally, interaction patterns among descriptors (e.g., between feature coverage and maximum inscribed circle radius) could be examined to understand their combined effects on bacterial attachment.

b) Normality Tests

Testing Assumptions for Predictive Models: Normality tests (such as the Shapiro-Wilk or Kolmogorov-Smirnov tests) would be useful for both the dependent variable (fluorescence intensity) and relevant topographical descriptors. Normality is often an assumption for metrics used in linear regression or for error analysis in some machine learning methods. Identifying non-normality would justify the use of transformations, which may improve the fit and reliability of models, especially for models like MLREM (Multiple Linear Regression with Expectation Maximization) that may rely on normally distributed residuals. Moreover, some machine learning models can be sensitive to highly skewed distributions. If normality tests indicate significant skewness in any descriptor, transformations (such as log, square root, or Box-Cox transformations) could standardize the data and improve model accuracy.

Testing Dependent Variable (Fluorescence Intensity) Normality: Since the study uses fluorescence intensity as the dependent variable, confirming its normality could help in selecting the appropriate performance metric and in interpreting the results. For example, if intensity values are skewed, the model's error metrics (e.g., RMSE) may need adjustments to reflect a more accurate predictive performance.

Improving Interpretability in SHAP Analysis: Testing for normality can ensure that the SHAP (SHapley Additive exPlanations) results reflect a balanced influence of descriptors. When features are normally distributed, SHAP values can provide clearer, more reliable insights into feature importance, which is beneficial for understanding the specific topographical traits that affect bacterial attachment.

c) Homoscedasticity Test

Checking for Equal Variance (Homoscedasticity): Homoscedasticity refers to the condition where the variability in the dependent variable (here, bacterial attachment levels) is constant across levels of an independent variable. In predictive modeling, heteroscedasticity (unequal variance) can undermine model performance by over-weighting data points with larger variances and under-weighting those with smaller ones. Testing for homoscedasticity (e.g., using Breusch-Pagan or White's test) could identify areas where variance needs stabilization. This may lead to adjustments in model error weighting or prompt the use of weighted regression techniques, thereby improving prediction accuracy.

Assumption Checking for MLREM and Random Forest Models: In linear regression models like MLREM, homoscedasticity is typically assumed. Detecting heteroscedasticity would highlight the need for alternative modeling strategies or for variance-stabilizing transformations on descriptors, potentially improving the model's linear fit. For Random Forests, while homoscedasticity is not a strict requirement, addressing heteroscedasticity can improve the reliability of performance metrics, especially in data subsets with distinct attachment patterns.

d) Implications for Model Selection and Performance

Guiding Model Choice: Normality and homoscedasticity tests help reveal the underlying structure of the data, which could inform model choice. For instance, if homoscedasticity is violated, models that handle variance better (like gradient boosting or generalized linear models) may be more suitable than linear regression approaches.

Optimizing Transformations: If exploratory analysis reveals non-normal or heteroscedastic patterns, transformations could be applied that make the data better suited to the selected models, potentially improving R^2 scores, error metrics, and interpretability.

3. Ambiguity in Data Representation: The fluorescence values are normalized to account for staining intensity differences. However, details about the normalization process (e.g., formulas or reference standards) are absent, compromising the reproducibility of the method. This step is crucial because any variation in staining intensity between experiments could introduce bias in the model predictions.

4. Model Validation: Although model robustness is implied by R^2 values and RMSE, the text could benefit from discussing alternative validation metrics which might provide a more comprehensive understanding of model performance on imbalanced datasets. Including also confidence intervals for these metrics, or presenting their variability across different bootstrap samples, would provide a more nuanced view of model performance.
5. Statistical Reporting and Interpretation: Some results, such as the average RMSE and R^2 values, are reported without discussing their potential variability or biological relevance. While high R^2 values imply good fit, additional interpretation on how this accuracy translates to practical, biologically meaningful outcomes would strengthen the study's utility.
6. Integration of ERI Model Comparison: While the study references Schumacher's Engineered Roughness Index (ERI), the comparison to the ML model lacks depth. Discussing how the ERI and ML models align or diverge in identifying essential descriptors would enhance understanding of these models' relative efficacy and limitations.
7. Limitations: The study provides valuable insights into bacterial attachment on specific topographies of polystyrene surfaces, but the findings may not automatically apply to other materials, complex surface designs, or environmental conditions (humidity, nutrient availability, and flow conditions). State clearly those limitations and expanding the range of topographies and materials in future datasets, combined with validation under diverse conditions, would be necessary to make the model's predictions broadly applicable across various real-world applications.

Reviewer #5

(Remarks to the Author)

Reviewer #6

(Remarks to the Author)

Authors have successfully addressed the majority of Reviewer 3s comments. However confusion still exists around the in vivo data and potential over interpretation of the host cell colonization data. Specific comments are below

- Labels in figure 2 are too small to legible
- Figure 3 and 4. What were the data normalised to?
- Figure 4c. Providing increased magnification images for 881Anti and 685Anti may be beneficial to more easily observe the biofilm formation on these materials. Do these surfaces affect the cell morphology?
- Figure 8. Overall magnification of the images are too low to be able to interpret them. The insets in (a) provide little clarification
- Figure 8b. Is the quantification of bacteria on the flat surface from the underside as in c-e, or were these separate controls. This was unclear
- Figure 8. Why are the surface topographies different for 881 between panels a and c?
- The host cell data from the mouse study seems to be over interpreted. Authors claim that host cell colonization of the implant helped prevent bacterial attachment. However there seems to be more host cell colonization on the Pro697 implant. Further there is no quantification of this data. Figure 8 is therefore difficult to interpret

Version 2:

Reviewer comments:

Reviewer #1

(Remarks to the Author)

The response to my comments is acceptable. I do not have further comments.

(Remarks on code availability)

N/A

Reviewer #4

(Remarks to the Author)

The revisions address all of the points raised. The manuscript is now acceptable for publication.

(Remarks on code availability)

N/A

Reviewer #5

(Remarks to the Author)

(Remarks on code availability)

Reviewer #6

(Remarks to the Author)

Authors have successfully address my previous comments

(Remarks on code availability)

Nature Communications Responses to Reviewers of NCOMMS-22-16395-T

Reviewer #1 (Remarks to the Author):

In this study, the authors screened a library of um size patterns and used machine learning to generate design rules for anti-biofilm surfaces. The work addresses an important problem and it is plausible to use ML in this area. However, the results contradict the general observations by other labs and did not investigate the mechanism. Thus, this reviewer felt it is pre-mature to publish in Nature Communications.

We believe that this is the first time a systematic approach employing machine learning has been applied to in the context of micro-topographies and anti-biofilm surfaces. We have very extensively revised our manuscript, fully placed the work in context with the literature and provided substantial new experimentally derived, mechanistic insights, into the biofilm resistance of the anti-attachment micro-topographies.

*In terms of biological mechanism, we have included the following new observations. We used single cell 3D tracking to visualize the motile behaviour of *Pseudomonas aeruginosa* on the micro-topographies. This revealed markedly different motility over flat surfaces and pro-attachment micro-topographies when compared with anti-attachment micro-topographies on which the bacterial cells swam along the channels and became entrapped in the crevices associated with the specific features. This finding led us to investigate the anti-biofilm mechanism involved. Neither the deletion of key genes involved in motility and surface sensing (*fliC* and *pilA*) and signalling (*wspF*) nor the constitutive expression of the cyclic diguanylate synthase gene (*yedQ*) facilitated biofilm formation on the anti-attachment micro-topographies. Instead, we discovered that activation of the Rhl quorum sensing (QS) pathway resulting in the production of anti-adhesive, lubricating rhamnolipid biosurfactants prevented biofilm formation on an anti-attachment microtopography. Mutation of the rhamnolipid biosynthesis gene, *rhlA* or the QS signal synthase gene, *rhlI* or the QS response regulator gene, *rhlR* enabled *P. aeruginosa* to form biofilms on the resistant micro-topographies. Conversely provision of either exogenous rhamnolipids or the restoration of quorum sensing in a *rhlI* mutant (by the provision of the Rhl quorum sensing signal molecule, N-butanoyl-homoserine lactone (C4-HSL)) inhibited biofilm formation on the resistant micro-topography.*

*These data are consistent with bacterial cell confinement in the anti-attachment feature crevices that results in auto-lubrication of the surface via rhamnolipid biosurfactant production so preventing biofilm development. Since rhamnolipid production is regulated by QS at high bacterial population densities in liquid cultures, this suggested that the likely restricted diffusion of QS signal molecules in *P. aeruginosa* trapped in the TopoUnit feature crevice microenvironment resulted in the induction of QS in a low population density leading to rhamnolipid production, so inhibiting biofilm formation. This is because the size of a 'quorum' is not fixed but depends on the relative rates of production and loss of the cognate QS signal molecule. In complex topographies containing cracks and crevices⁵⁷ or where bacteria become trapped in vesicles⁶⁷, QS signal molecules can accumulate and activate target gene expression at lower population densities i.e. *P. aeruginosa* cells trapped in the TopoUnit feature crevices can become quorate.*

Specific comments:

The results show that less cells attach to the surfaces with narrowly spaced patterns. As the authors acknowledged, this is the opposite of other studies. In addition to refs. 18 and 19

mentioned, Ref 30 (Friedlander et al.) cited in this paper showed how bacterial flagella interact with topographical features and the role of pattern. The results showed closely positioned pillars promote biofilm formation, unlike reported here. While it is good to present different results, the authors did not offer an explanation or speculation about the possible mechanism/cause of the differences. For example, is that possible that some air was trapped between the close features and thus led to the Wenzel state and inhibited bacterial attachment? If not, was there any treatment of the samples to prevent that?

There are indeed contradictory publications which are often a consequence of using roughness as a simple descriptor for surface topography. As Cheng et al (revised version ref. 28) highlight, surfaces with drastically different topographies can have similar roughness parameters. Roughness describes only the height variation of a surface, whereas topography represents the configuration of a surface in a three-dimensional space, often characterized by vertical features (e.g., protrusions and recessions) and their spatial arrangement. Being an amplitude parameter, roughness does not capture any spatial information including the geometric details density, periodicity, symmetry or hierarchical arrangement of surface details, many of which have been shown to play a critical role in bacterial attachment. We have discussed these issues in the revised version (revised version, page 5 lines 127-137).

*Furthermore, our results may differ from other publications because the feature dimensions and channels are different. For example, in the Friedlander paper revised version Ref 58) the narrow channels used in their hexagonal unit array were mainly 440 nm in width, with hexagon features that were 2.7 μm in height and 3 μm in width and promoted *E. coli* biofilm formation after an initial reduction in adhesion during the first 2 h. These channels are likely too narrow for *E. coli* cells to swim along. Our TopoUnit 881 channels were 2 μm which we show were readily accessed by *P. aeruginosa* (revised version Fig. 5 and Video S1) and incorporated much taller feature pillars (10 μm). We have now presented a detailed mechanism to explain the *P. aeruginosa* biofilm resistance of our anti-attachment micro-topography (revised version pages 13 lines 336 to page 15 line 395 and page 19 line 514 to page 20 line 544). In addition, as our new data show (revised version Fig.5 and Video S1) *P. aeruginosa* rapidly swimming along the channels in 881. Therefore, there is no trapped air preventing bacterial cell access to the channels. Any trapped air was removed prior to inoculation - this is now described in the Methods: revised version on page 24 lines 653-658, we have added:*

“It should be noted that despite air bubbles forming on the Topochip surface after media immersion, air pockets were removed by brief incubation at 37°C and repeated media pipetting until no entrapped bubbles between the topographical features could be detected in the experiments carried out. In addition, for narrowly spaced TopoUnit patterns such as 881, individual bacterial cells were clearly be observed swimming along the 2 μm channels between the pillars.

The in vivo study gave 4 days between implantation and bacterial inoculation. This allows the host cells to cover the surface and thus it is not clear if the inoculated bacteria can still have access to the patterns. A shorter gap can help better see the effects.

We wanted to mimic the clinical implantation and subsequent medical device infection to determine the combined impact of host and implant microtopography collectively on infection. The pro-attachment data clearly demonstrate that the bacteria still have access to the patterns after 4 days.

The following has been added to the revised version, page 15 lines 413-415: ‘This method enabled us to mimic implantation and colonization of a medical device and to evaluate the combined impact of host and micro-topography on the course of infection’. Also, on page 16 lines 424-428 ‘Since the TopoChips were implanted 4 days prior to infection, it is likely that host cell colonization of the

implants contributed to the reduction in bacterial attachment, in essence winning the 'race for the surface' (Gristina et al., 1988). However, pro-attachment 697 TopoUnit data clearly demonstrate that bacteria maintain access to the micro-topographies.

Some literature citations are incorrect. For example, Ref. 30 has um size patterns, not nanopatterns as the authors claimed. There are also other studies on topographical features not cited here. It is impractical to cite all, but giving a better covered Introduction can help the audience understand the field.

Ref 30 (Friedlander – revised version, ref 58) mostly has 440 nm channels in the Hex arrays used although some experiments with channels up to 1.7 μM were included. We described Sharklet AF which has micro-topographical features plus many other references including reviews that contain tabulated summaries of papers on topographies and bacterial biofilms. In the revised version, see the references 25, 29,31,35,36 and 37.

It is surprising that, although different shapes were tested, only the descriptors of circles are among the top 5 identified. Some explanation/speculation would help.

We have revised the relevant text (see revised version page 8 line 208 to page 9 line 245). We also provided additional information on the ML analysis and the method for descriptor selection (Text S1). Table S1 provides an explanation of all 68 descriptors evaluated.

P7, 18. The authors claimed that their new patterns are better than Sharklet. This direct comparison is inappropriate since the conditions are different. The cited test of Sharklet was 4 days in rich media and 24 h in minimal media, while this study tested 4 hour attachment in a rich medium. In addition, the ref did report Sharklet biofilm control for both MASA and P. aeruginosa. The statement that Pa results of Sharklet are not available is incorrect.

We have deleted the text directly comparing our results with Sharklet in our revised version. However, we disagree that our statement on P. aeruginosa was incorrect. We stated that comparable wild- type P. aeruginosa data were not available because in the Ref 15 (ref 25, revised version), the authors' wild type P. aeruginosa was stated not to form a robust biofilm on Sharklet under the conditions used. Instead they used a P. aeruginosa bifA mutant.

The two most effective designs (881 and 685) are only shown in a small image in Figure 4. The actual dimensions and possible mechanism of biofilm reduction should be described. It is also important to describe the detailed results of other similar patterns (same size but different shape/spacing or same spacing but different size/shape).

*Expanded SEM images of anti-attachment Topo-units 881 and 685 as well as 632 and 1497 are/were shown in Fig. S2a (original and revised versions). In addition, enlarged images for pro-attachment Topounits 697, 336, 1556 and 1527 were/are shown in Fig. S2a. The detailed data was/is presented as intensity maps and fold changes in attachment compared with flat surfaces for multiple pathogens on each of these different Topo-units - see Fig. S2b. The dimensions of the primitives, features and their spacing were shown in Fig S1a. **Additional details are given in the Results, revised version page 7 lines 174-183 and in Figs. S1 and S2).** The biofilm inhibition mechanism has been investigated for P. aeruginosa and extensive new data included in **the revised version pages 13 lines 336 to page 15 line 395 and page 19 line 503 to page 20, line 544)***

The video links do not connect to the specific video files but rather the website of the team.

The videos in the original version have been replaced with new videos S1-S3 which have been uploaded onto the journal submission site.

P11, 24-26. This statement is not true. For example, <https://www.nature.com/articles/s41551-021-00739-4>.

We disagree that our statement above is incorrect. The paper referred to by the reviewer is Doloff et al, *Nature Biomed Engineering*. It describes breast implants with different irregular surface roughnesses (average roughness, 0–90 µm) that do not possess topographical surface features with well-defined dimensions and shapes as presented by the Topounits.

Our statement (revised version page 16 lines 440-444) now reads: The importance of the design of topographically patterned polymers with well-defined dimensions and shapes has previously been noted in vitro in terms of macrophage recruitment (ref 23). Therefore, the intrinsic biofilm resistant properties of anti-attachment TopoUnits such as 881 combined with a productive host response could potentially play a role in reducing implanted medical device-associated infections’.

The resolution of Fig. 2 is low.

It appears to be OK on our screens.

L11, 10-12. It is unclear how this statement can be made since the surface is already covered by host proteins and cells.

The statement was: ‘Since bacterial attachment was prevented by anti-biofilm topographies in vitro in the absence of host cells, the intrinsic biofilm-resistant properties of specific Topounits such as 881 clearly play a role.

We have modified the statement to: “Therefore the intrinsic biofilm-resistant properties of anti-attachment Topounits, such as 881, combined with the host response could potentially play a role in reducing implanted medical device associated infections.” Revised version Page 16 lines 442-444.

Reviewer #2 (Remarks to the Author):

The paper proposed polymeric surface patterns that dramatically limit bacterial pathogens adhesion. By using a high-throughput unbiased screen of micro-topographical patterns on polystyrene derived from the assembly of primitive designs, authors identified specific patterns that significantly prevent the adhesion of 2 important bacterial pathogens (*Pseudomonas aeruginosa* and *Staphylococcus aureus*). Despite important differences between these two species (motility, cell morphology, cell wall decorations...) the correlation between the behaviour of these two species suggests the robustness of the antiadhesive effect. This robustness was extended with different materials, two other bacterial species, in an in vitro biofilm assay and finally in a murine model. A machine learning approach allows pinpointing specific traits of antiadhesive patterns that could be integrated into the development of bioactive surfaces. The general strategy and the efficacy of the micro-patterned antiadhesive surfaces are impressive. The paper is well constructed, and well written and the presented data are in accordance with the conclusions.

We thank the reviewer for their positive comments.

I have however a few concerns with the article:

-in some figures, while the fluorescence images presented suggest almost no bacterial cells on the antiadhesive micro-patterned surfaces compared to the flat control, the quantification of the

reduction appeared less contrasted. For example, on page 8 line 13 it is mentioned a 5-fold average reduction for *S. aureus* and in figure 4C the images presented suggest a higher level of reduction (the biofilm is almost covering the control and no cells are visible on the anti-adhesives surface). Is it associated with the selection of the presented images? Or to variation of fluorescence intensity on the different materials? In that case, could a segmentation step before the quantification (determination of the biovolume) be relevant?

The 2000+ TopoUnits on each Topochip present several challenges for confocal fluorescence microscopy compared with flat surfaces. These include the background fluorescence which not only differs between experiments and between topographies due to lensing effects caused by the polystyrene and the different shapes of the features. Hence, we tried a number of different approaches and found that best and most reproducible way to quantify fluorescence intensity on the Topochips was to use a thresholding strategy that involved segmentation step. This latter was missing from the original text which now states: *“assay-specific background removal by applying the automatic thresholding function of Fiji-ImageJ to flat control images and the average minimum threshold value obtained was used as baseline to segment all images from the same experiment”*. **(Revised version page 25 lines 687-689)**. With respect to Fig. 4C, the single images selected for *Staph. aureus* are not only representative but there is clearly some cell attachment visible on the two anti-attachment surfaces in the originals such that at least a 5 fold difference from multiple images would not in our view be unreasonable.

- Page 8 line 29 and figure S5: what dyes were used for the live/dead staining, syto9 and propidium iodide (PI) proposed in the live/dead labelling? The mean fluorescence intensity presented in the control surface is very similar for the “live” and “dead” populations. How were these populations calculated: while PI is labelling only the dead cell with compromised integrity of their membranes, Syto 9 is labelling both live and dead populations. Quantification of live and dead populations is not directly the fluorescence signal of syto 9 and IP. Authors should clarify, either by explaining how they extracted the live and dead proportions or by presenting the raw signals with Syto9 and IP (and not extrapolating the relation with live and dead cells in the figure). Then, if 50% of the initial adherent population is dead on the controls as suggested here, the authors should discuss this unexpected observation.

The BacLight viability kit we described uses Syto9 and propidium iodide. However, we have removed the live dead staining from the revised manuscript (a) because of the difficulty of obtaining a robust live:dead ratio with so few cells on the anti-attachment surfaces and (b) because of the extensive new mechanistic dataset we have now included.

-The polymeric chip design and the general strategy are very similar to reference 17 published in PNAS in 2011. The main difference is the target cells (eucaryotic cells in the PNAS paper, bacteria here). While the results obtained with bacteria are impressive, this lowers a bit the novelty of the approach to generate those patterns.

With respect, bacteria are very different to eukaryotic cells which don't form biofilms. Bacteria are also very much smaller and there is no commonality with the osteogenic differentiation and proliferation of mesenchymal stromal cells in response to specific TopoUnits investigated in revised version ref. 15.

-Page 4 line 27: *...” early cell attachment could be prevented and AND HENCE BIOFILM FORMATION after longer incubation times”*. Please modulate this sentence: there is no universal correlation between the number of adherent cells and the biomass of the mature biofilm.

*While we agree that there is no universal correlation between the number of adherent cells and the biomass of the mature biofilm, if bacteria do not attach to a surface, then they cannot form a biofilm on that surface. Nevertheless, we have modified the sentence in the **revised version (page 8, lines 201-203)** to ‘These findings strongly suggested that by modifying surfaces with well-defined micro-topographies, cell attachment and subsequent biofilm development could be inhibited’*

-Minor comments:

-On page 4 line 13, remove the symbol between “Pseudomonas” and “aeruginosa”

Corrected

-On page 11 line 17, remove the symbol between (top) and “and flat”

Corrected

-page 20 figure 4D: the choice of the scale of the 2 Y-axis appears arbitrary; could the authors find a way (e.g. normalize the fluorescence on the control flat surface) to compare the 2 strains?

*In the **revised version**, we have normalized the data in **Fig. 4d** and all the other figures (**Fig. 4b, Fig. 3b, Fig 6 (b and d), fig 7 (b and d)**) reporting quantitative fluorescence intensity.*

-Page 22 Figure 6a: Here also, the images presented are not in concordance with the quantitative data presented in Figure 6b (the pro surface 697 appears far less contaminated than the flat control on the fluorescent and not on the average histogram). images while the quantification.

*The single fluorescent images shown in Fig. 6a (**revised version, Fig. 8a**) for flat, 697 and 881 implants recovered from the infected mice are simply representative images. The quantitative data showing the scatter from multiple implants were presented in Fig. 6b (**revised version, Fig. 8b**)*

Reviewer #3 (Remarks to the Author):

Overall, the idea of screening topographical designs computationally is very interesting, but limited to known descriptors. Hence, I am not convinced that bioinstructive materials, as presented here, have a real-life impact to reduce the burden of medical device-associated infections (as this was the focus that the authors have chosen). The materials reduced adherence but not biofilms entirely and their ideas seem to be preliminary at this point as long term studies are missing – it will be important to see whether the material is indeed of practical use. After reading this manuscript, I think the real-world application of improved materials is rather in the industry (e.g., biofouling) than in the health sector, as it could reduce cleaning costs / frequencies etc.

While we agree with the reviewer that the work presented is of broad relevance wherever there is a problem with surface associated biofilms, we disagree that the anti-attachment micro-topographies identified here are not relevant to medical device-associated infections. These are a major unmet medical need - as yet no textured devices have made it to the clinic.

*We believe that this is the first time a high throughput systematic screening approach coupled with machine learning has been applied to in the context of micro-topographies and anti-biofilm surfaces. While the work may not yet be translation ready, we have identified novel biofilm inhibitory micro-topographies and in the revised version (see response to reviewer 1), an extensive additional dataset on the molecular mechanism by which biofilm formation by *P. aeruginosa* on these topographies is blocked. In the context of infection, we also show that our lead micro-topography also impacts productively on the host response to infection. Preventing biofilm formation also renders the*

infesting bacteria easier for host phagocytic cells to clear from an implanted device than mature biofilms which are much more refractory to host immune clearance.

A 15-fold reduced colonization is not striking (although it might actually be higher depending on how to interpret the data). The mentioning of motile and non-motile bacteria should be rephrased, as there is enough evidence that *S. aureus* performs certain motility behaviours

*We disagree that a 15 fold reduction is not striking – the changes relate to biofilm biomass not viable counts. In addition, S. aureus is not a classically motile bacterium. It lacks the flagella and retractable pili required for swimming, swarming and twitching motility. There is a report of a S. aureus colony spreading via surfactant mediated ‘sliding’ on agar surfaces. We have changed non-motile to ‘non-flagellated’ in the **revised version (p8 line 199)**.*

and *P. aeruginosa* would prefer to attach to an air-liquid interface (which was not studied) rather than submerged material (where gravity would pull them down anyways). Also, *P. aeruginosa* is usually not a good biofilm-former in nutrient rich broth such as TSB (as bacteria naturally detach after a certain amount of time); which should be considered when interpreting the overall significance of the data.

Biofilm formation by P. aeruginosa is highly dependent on the strain and the growth environment used and forms biofilms on both submerged surfaces and at air-liquid interfaces both of which are encountered in real world situations. We disagree with the generalization that P. aeruginosa prefers to attach to an air-liquid interface.

We used TSB so that each of the 4 bacterial species could be cultured in the same growth medium. For some experiments, TSB supplemented with 10% v/v human serum (TSB HS10%) to mimic in vivo conditions was used. The initial read-out for our studies was not biofilm formation but attachment after 4 h. The PAO1 strain used in our experiments formed robust biofilms after 24 h growth in TSB. It should also be noted that PAO1 sub-lines from different labs do show growth medium dependent differences in biofilm formation.

*In the revised version (**page 20 lines 570-571**) we have explained our choice of growth medium as ‘In order to maintain the same culture conditions for each bacterial species, tryptic soy broth was used as the growth medium for all Topochip assays’ (revised version lines 570-572). ‘We have also added (**revised version page 11, lines 293-295**) that “Mature biofilms incorporating interconnecting bacterial aggregates were clearly observed on the pro-attachment TopoUnits 697 and 336, as well as on the flat control, while the few cells present on the anti-attachment TopoUnits 881 and 685 were far more dispersed.”*

Moreover, the experimental setup for biofilm attachment is somewhat problematic as well, as diluted overnight cultures were used. For example, if *P. aeruginosa* overnight cultures were in late stationary phase, there is a very high chance that they won’t express flagella (which would be required for immediate attachment in the new growth conditions). In this regard, the rationale of flagella and pilus mutant experiments are rather weak as they might not even be present in the wild type. In addition, one would at least expect to have complementation studies as well.

*Our starting inoculum was always an overnight culture, diluted into fresh TSB to an OD₆₀ 0.1 and grown for 4 h such that *P. aeruginosa* was in early log phase and expressing flagella. Swimming motility is clearly demonstrated in the tracking data and videos (**see revised version Fig. 5, Fig. S6 and videos S1-S3**). In the **revised version page 24 lines 663 to 665**, we now state: The air-dried chips were placed in petri dishes (60mm x 13mm) and incubated statically at 37°C in 10 ml of growth*

medium inoculated with actively growing bacteria for 4 h or 24h (from optical density: $OD_{600\text{ nm}} = 0.01$).

Our mutants have been genetically complemented. For *pilA*, *fliC*, *rhIA*, *rhIR* and *rhII*, see **revised version Table S2 and Fig. 7 and Fig. S7**.

But, I would have appreciated if they authors would have shown a biofilm overproducer strain rather than these ko-mutants.

Give the importance of pili and flagellar in surface sensing in early stage attachment, we felt it was important to include these T4 pilus and flagella mutants. There is no obvious candidate target gene for biofilm 'overproducer' in P. aeruginosa. However, cyclic diguanylate signalling controls the switch in P. aeruginosa from motile to biofilm lifestyles and can be induced either by the introduction of a cyclic diguanylate synthase (e.g. yedQ from E. coli; Chen et al 2015) or mutation of wspF (Hickman et al 2005). We have included new data with both of these strains and they do not promote biofilm formation on the anti-attachment microtopography (revised version Fig. 6c and 6d; page 13 lines 354-365).

The other missing link would be experiments with increasing cell numbers; what happens if more bacteria are used – does this affect attachment? I really would have appreciated bacterial counts to confirm the presence (or absence) of bacteria. The quantitative representation is not providing the full picture.

It is difficult to remove all the bacteria attached to the micro-topographies to obtain meaningful viable counts. The confocal microscopy methodology used is accepted as standard in biofilm research as it allows quantification of biomass and other parameters. Higher viable counts do not necessarily equate with greater biofilm biomass. We therefore do not think that trying to increase inoculum size or doing viable counts will add new insights into the mechanism involved especially given the new biological mechanistic data provided in the revised version (See response to reviewer 1).

The in vivo experiments are somewhat confusing as well. Figure 6b shows individual explants – does this mean individual mice? What was the power calculation for these experiments? It almost appears that data was 'forced' to be 'significant' as more than twice the amount of explants were used for the control vs Topunits.

We have clarified the in vivo experiments in the revised manuscript. See revised version page 15, line 406-409, page 27 lines 746 and page 41, Figure 8 legend. Each mouse received 1 implant and 8 mice were used (i.e. n=8) per implant type (flat, pro (697) and anti (881)) – 24 mice in total. For each recovered implant, 4 random images of the surface (revised version page 28 lines 762-763) were captured for quantification of bacterial colonization and immunohistochemistry.

Since the P. aeruginosa strain PAO1 was used in the in vivo studies I was wondering about the mortality rate and how often did the mice cleared the infection?

Our UK government home office animal experimentation licence does not permit us to allow infected mice to die of infection. In addition, the mice were sacrificed 4 days post-infection so that we could recover the implants. We were therefore not able to follow clearance of the infection. As the revised manuscript contains a substantial number of new experimental findings, we feel that additional animal experiments are well beyond the scope of this manuscript.

Minor:

Figure 4 does not show studies with Sa as indicated in the legend and is missing complementation studies. The annotations in the figure are also wrong as the delta sign is missing

*We disagree. **Figs 4c and 4d** do show data for *S. aureus*. This was a wild type *S. aureus* strain so nothing to genetically complement. No mutant data for *S. aureus* are shown in Fig. 4. The figure legend is correct.*

Responses to Reviewers Comments

We first address the biological (Reviewers 1, 2 and 6) prior to the machine learning remarks (Reviewers 4 and 5)

Reviewer #1 (Remarks to the Author):

The authors conducted additional experiments and significantly revised the manuscript. I have a few additional comments:

Line 346. Is this decrease statistically significant? If so, the bar graph should show the * signs of related comparisons and the strains of genetic complementation should also be tested.

Fig. 6 (b) has now been revised as by including with additional * symbols. However, the loss by mutation of type IV pili (*pilA* deletion) or flagella (*fliC* deletion) makes no difference to the ability of *P. aeruginosa* to form a biofilm on the anti-attachment Topo-Unit 881. However, there were no obvious differences between the $\Delta pilA$, $\Delta fliC$ mutants and the parent strain with respect to the anti-attachment TopoUnits. This we had stated on **lines 500-502** in the Discussion section.

We genetically complemented the strains to demonstrate that twitching and swimming motility was restored in the mutants (**Fig. S8 (c)**). We therefore do not consider that it is necessary to run additional Topounit assays since the loss of swimming or twitching did not contribute to the differential attachment of *P. aeruginosa* to 881. We subsequently provide evidence for a mechanism that involves lubrication via quorum sensing-dependent rhamnolipid production. However, to avoid any confusion, we have deleted the sentence on **lines 345 and 346 (original version)** which stated 'However both mutants showed reduced levels of attachment to both flat and pro-attachment Topounits (**Fig. 6(b)**)'.

Line 359. Has the timing of c-di-GMP production in the WT strain been considered? To compare with the results of mutants, it is important to know the level of c-di-GMP in the WT under the experimental condition of this study.

The deletion of *wspF* or constitutive expression of *yedQ* both result in the constitutive overproduction of c-di-GMP. However, biofilm formation on the anti-attachment TopoUnit 881 was not restored. Therefore, the mechanism by which Topounit 881 inhibits biofilm formation is unlikely to involve the c-di-GMP driven switch from the motile to the biofilm lifestyle and must depend on an alternative mechanism as we go on to demonstrate. Investigating the timing of c-diGMP production given our subsequent findings is unlikely to provide any further mechanistic insights.

There is some extensive discussion about the possible role of QS in the observed effects on attachment. It would be helpful to test a QS reporter strain to see if the trapped cells indeed have a higher level of QS activities.

We have provided a very extensive dataset on the contribution of QS-driven rhamnolipid production by

deleting (i) the rhamnolipid biosynthesis gene *rhIA*, (ii) the C4-HSL synthase gene, *rhII* and (iii) the C4-HSL receptor gene, *rhIR*. All three mutants were genetically complemented and we also added back exogenous C4-HSL to the *rhII* mutant which restored rhamnolipid production and inhibited biofilm formation. These data unequivocally demonstrate a role for QS and rhamnolipids in the biofilm resistance of Topo-Unit 881. Although a QS reporter strain might possibly provide some nice additional images, we do not have a reporter that is sufficiently sensitive to detect C4-HSL production by a few trapped cells within the anti-attachment topographies.

Reviewer #2 (Remarks to the Author):

The authors have adequately addressed my comments, and the article is now in an appropriate form for publication.

Reviewer #6 (Remarks to the Author):

Authors have successfully addressed the majority of Reviewer 6’s comments. However, confusion still exists around the in vivo data and potential over interpretation of the host cell colonization data. Specific comments are below

Labels in figure 2 are too small to legible

We have increased the label font size in Fig. 2

Figure 3 and 4. What were the data normalised to?

We explained the normalization in the Methods section, **lines 704-707 (original version)**. The fluorescence data in **Figures 3 and 4** corresponds to the fluorescence values for bacterial attachment to each of the replicate TopoUnits normalised to the average fluorescence intensity of the chip. This was to account for any differences in staining intensities between experiments.

Figure 4c. Providing increased magnification images for 881Anti and 685Anti may be beneficial to more easily observe the biofilm formation on these materials. Do these surfaces affect the cell morphology?

We have included a new figure (**Fig. S6**) in the **Supplementary Information** showing environmental scanning electron microscopy (ESEM) images of *P. aeruginosa* on (a) flat, (b) pro-attachment (TopoUnit 697) and (c) anti-attachment (TopoUnit 881). No obvious differences in cell morphology were apparent.

Figure S6. Environmental scanning electron microscopy (ESEM) showing the morphologies of single *P. aeruginosa* cells on (a) flat, (b) pro-attachment (TopoUnit 697) and (c) anti-attachment (TopoUnit 881) micro-topographies after 4 h incubation. Scale bars 10 µm.

- Figure 8. Overall magnification of the images are too low to be able to interpret them. The insets in (a) provide little clarification

With respect to **Fig. 8**, we have removed the small low resolution inserts from panel (a) and replaced them with separate higher resolution panels (bottom panels below) where fluorescent and brightfield images have been overlaid.

- Figure 8b. on the flat in c-e, or were these separate controls. This was unclear

Is the quantification of bacteria surface from the underside as

As stated in the methods (**lines 745 to746; original version**) and **Fig.8 (b)**, we implanted either Topounits or flat polyurethane samples of the same dimensions. We have now further clarified this in the revised version methods (**lines 761-762, revised version**), results and in the figure legend for panel **8(b)**.

- Figure 8. Why are the surface topographies different for 881 between panels a and c?

The 881 surface micro-topographies in panels 8(a) and 8(c) were the same although the insert resolution was poor and there was a scale bar error. As above, in the revised version, we have included higher resolution images – see **Fig. 8(a)** lower panel.

The host cell data from the mouse study seems to be over interpreted. Authors claim that host cell colonization of the implant helped prevent bacterial attachment. However there seems to be more host cell colonization on the Pro697 implant. Further there is no quantification of this data. Figure 8 is therefore difficult to interpret

The importance of **Fig. 8** is that it demonstrates that TopoUnit 881 maintains its anti-attachment properties *in vivo* in a mouse infection model. Since much more work will be required to establish clearly the precise contribution of the host immune defences to attachment resistance of micro-topographies such as 881, we have modified/removed the text with respect to ‘productive host response’ in lines **53, 173, 578, (revised version)** to avoid over-interpretation. We have also further simplified **Fig. 8** by removing panel (e) and revising the text, replacing **lines 418-444** in the original version with **lines 341-445** (revised version):

‘Infected mice implanted with TopoUnit 881 showed significantly lower *P. aeruginosa* pcE2C colonization compared with those receiving flat or TopoUnit 697 as determined using quantitative confocal fluorescence microscopy (**Figs. 8 (a) and (b)**). Antibody-based staining of *P. aeruginosa* on the explanted 881 and 697 TopoUnits confirmed these findings (**Fig. 8 (c)**). Since the TopoChips were implanted 4 days prior to infection, it is possible that host cell colonization of the implants contributed to the reduction in bacterial attachment, in essence winning the ‘race for the surface’. However, **Fig. 8(c)** clearly shows that for TopoUnit 697, both host cells and bacteria maintained access to the micro-topography. Furthermore, an asymmetric host response to the TopoUnits, illustrated by staining of a representative PU TopoUnit 881 recovered from an un-infected (PBS-inoculated) animal is shown in **Fig. 8 (d)**. Here, the implant was removed from the mouse tissues during sectioning, leaving a void shown as a black area in each image. Host responses to the micro-topographically patterned (top) and flat sides (bottom) were very different. Greater host cell migration into the TopoUnit side of the implant (indicated by the blue DAPI-stained nuclei) compared with the flat underside is clearly apparent (**Fig. 8d**). These data demonstrate that Topounit 881 maintained its colonization resistance properties *in vivo*’

Machine Learning Comments

Reviewers #4 and #5 (Remarks to the Author):

General Comments

The paper states that "Attempts to prevent such infections have predominantly focused on the incorporation of leachable bactericidal antimicrobials into medical devices"; this is not strictly the case, with significant work also being done into other methodologies with both bacteria and cells. This large body of work is also at odds with the statement "Mechanistic underpinnings to cellular responses to materials and their chemical properties are lacking, preventing reliable de novo prediction and design of improved materials". The use of cells, bacterial cells, and similar terminology adds to this unclear message, as the reader can be confused as to what specific previous research and known information is being referred to. With the above in mind, a stronger, clearer rationale for the use of machine learning would benefit the paper including what specific gaps in knowledge and research are being addressed, and how, as well as how this will translate into the field

Response to General Comments

In the revised version (**line 83**) we have modified the sentence to '*Attempts to prevent such infections have **often** focused on the incorporation of leachable bactericidal antimicrobials into medical devices*'.

While there is clearly a need for a much better molecular mechanistic understanding of the diverse interactions between microbial and mammalian cells, we have deleted the sentence '*Mechanistic underpinnings to cellular responses to materials and their chemical properties are lacking, preventing reliable de novo prediction and design of improved materials*'.

With respect to the rationale for using ML, we stated (**original version**) in the **Introduction lines (99-101)**:

Using machine learning, strongly predictive relationships linking physicochemical properties of materials to biological responses have subsequently been identified, allowing the design of polymer chemistries with greatly improved biofilm inhibitory properties^{20,21}.

Responses to Machine Learning Comments

1.Lack of Methodological Justification: There is not any justification why Random Forest or MLREM were used, other models like gradient boosting (e.g., XGBoost), SVMs, or neural networks could be more suited to the dataset's structure and the prediction task.

Moreover, the choice of Random Forest default parameters is stated but not justified. This omission limits reproducibility because default parameters may yield different results in other implementations or environments. Moreover, without information on parameter tuning or optimization, it's challenging to determine if these default settings are optimal for this dataset.

Response:

We tested multiple machine learning approaches in the dataset, as further detailed below. We decided to report the best machine learning (marginally) and the results of a linear model, as the beta coefficients were more easily interpretable, even though the data are not normally distributed. We also observed strong linear relationships between the main selected descriptors and bacterial attachment.

The use of standard parameters for the machine learning methods investigated was chosen as the dataset is small and has observed linear relationships between a few important independent variables and the dependent variable. Using the Python package described in the paper, one can have access to the parameters from the Python documentation and therefore reproduce the results. If using other environments, such as MATLAB or SPSS, it is likely that differences will occur even with the same use of hyperparameters, as the approach implementation will likely differ.

Therefore to address the reviewers' comments, we have added the following information to the main text (**lines 145-152** revised paper):

'Here, we describe the generation of predictive statistical and machine learning models for bacterial pathogen attachment using topographical descriptors that encode geometric elements. We used these models to Forest and Multiple Linear Regression with Expectation Maximisation results are reported after finding that the models generated using these were marginally superior to SVM and XGBoost models. The consistency found in results from the different modelling approaches enabled us to identify key feature shape parameters with the highest relevance for bacterial attachment'.

Due to the nature of the dataset and the stability of the results even with a simple linear model, it is unlikely that very significant differences will be observed when modelling the data using alternative non-linear ML approaches, as further demonstrated by the results below, for *S. aureus* and *P. aeruginosa*, with the same initial seed for randomly selected datasets, and 50 bootstraps. XGBoost yields good results, although it appears to overfit to the training data. The results are summarized as follows:

S.aureus

P.

In Table summary form for *S. aureus*:

Model	Set	MAE	R2	RMSE
Linear Model	Test	0.260 ± 0.009	0.704 ± 0.017	0.341 ± 0.013
Linear Model	Train	0.257 ± 0.004	0.709 ± 0.007	0.338 ± 0.006
Random Forest	Test	0.168 ± 0.005	0.858 ± 0.012	0.236 ± 0.010
Random Forest	Train	0.136 ± 0.002	0.918 ± 0.003	0.180 ± 0.003

SVM	Test	0.163 ± 0.005	0.858 ± 0.013	0.236 ± 0.012
SVM	Train	0.157 ± 0.002	0.867 ± 0.006	0.229 ± 0.005
XGBoost	Test	0.172 ± 0.005	0.848 ± 0.013	0.244 ± 0.011
XGBoost	Train	0.079 ± 0.002	0.971 ± 0.002	0.107 ± 0.003

In **Table** summary form for *P. aeruginosa*:

Model	Set	MAE	R2	RMSE
Linear Model	Test	0.195 ± 0.005	0.692 ± 0.031	0.256 ± 0.013
Linear Model	Train	0.194 ± 0.002	0.692 ± 0.013	0.256 ± 0.006
Random Forest	Test	0.148 ± 0.005	0.823 ± 0.015	0.194 ± 0.008
Random Forest	Train	0.124 ± 0.002	0.882 ± 0.004	0.159 ± 0.002
SVM	Test	0.143 ± 0.004	0.832 ± 0.012	0.189 ± 0.007
SVM	Train	0.135 ± 0.002	0.851 ± 0.005	0.178 ± 0.003
XGBoost	Test	0.153 ± 0.005	0.807 ± 0.023	0.202 ± 0.010
XGBoost	Train	0.075 ± 0.002	0.953 ± 0.002	0.100 ± 0.002

These steps are crucial and would enhance the study in several ways:

a) Exploratory Data Analysis (EDA)

Feature Distributions: EDA would allow researchers to understand the distribution of each topographical descriptor and fluorescence intensity variable. This could reveal patterns like skewness, outliers, or distinct clusters, which may affect model accuracy and interpretability. If skewed data is identified through EDA, transformations (e.g., log or square root) could be applied to stabilize variance and improve model predictions.

Responses:

Feature Distributions

We have conducted both clustering analysis and exploratory analysis of the main descriptors, as exemplified below. Results overall showed no significantly improved results when modelling the problem per cluster. Outliers were investigated and it was concluded that those were present due to manufacturing defects and that was the reason for the unexpected observed outcome. Exclusion of those outliers did not significantly change the model results, due to their scarcity in comparison to the whole data trends. All information on the detailed analysis has now been included in the **Supplementary Information**.

Study of the interplay between important descriptors:

The figure below shows the differences observed when investigating the data distribution for the descriptor Total Area and a composite variable resulting of the multiplication of the area times the average of the inscribed circles. As can be seen, the trends are similar. SHAP was further used to understand descriptor synergy.

Descriptor Relationships and Correlations: EDA could visualize correlations between descriptors, clarifying whether the cut-off threshold for removing highly correlated features (Pearson correlation > 0.85) is suitable. For instance, a lower or higher threshold may capture more relevant descriptors or remove noise, potentially increasing model robustness. Additionally, interaction patterns among descriptors (e.g., between feature

coverage and maximum inscribed circle radius) could be examined to understand their combined effects on bacterial attachment.

Response:

The feature selection approach using SHAP takes into consideration independent variable importance, but also the synergy between those variables. This was used as a second pass for data dimensionality reduction (in addition to correlations and careful removal from experimentalists), ensuring that the remaining variables were relevant to the problem and interpretable, both individually and combined.

b) Normality Tests: Testing Assumptions for Predictive Models: Normality tests (such as the Shapiro-Wilk or Kolmogorov-Smirnov tests) would be useful for both the dependent variable (fluorescence intensity) and relevant topographical descriptors. Normality is often an assumption for metrics used in linear regression or for error analysis in some machine learning methods. Identifying non-normality would justify the use of transformations, which may improve the fit and reliability of models, especially for models like MLREM (Multiple Linear Regression with Expectation Maximization) that may rely on normally distributed residuals. Moreover, some machine learning models can be sensitive to highly skewed distributions. If normality tests indicate significant skewness in any descriptor, transformations (such as log, square root, or Box-Cox transformations) could standardize the data and improve model accuracy.

Response:

The results of both normality tests suggest that all variables, including both the dependent and independent variables, **do not follow a normal distribution**. This non-normality has implications for the choice of statistical or machine learning models, particularly those that assume normality (e.g., linear regression, parametric tests). For our analysis we have normalised the data before modelling and chosen non-linear ML approaches.

Normality tests for *S. aureus*:

Variable	Shapiro-Wilk Statistic	Shapiro-Wilk p-value	Kolmogorov-Smirnov Statistic	Kolmogorov-Smirnov p-value
Total Area	0.994	0	0.022	0.23
Average Inscribed Circle Radius	0.96	0	0.044	0
Maximum Inscribed Circle Radius	0.947	0	0.1	0
Maximum Feature Radius	0.901	0	0.122	0
Inscribed Circle Radius 0.9 Percentile	0.944	0	0.08	0
Average Fluorescence S. aureus	0.969	0	0.062	0

Dependent variable distribution for *S. aureus*

The **Shapiro-Wilk test** for normality on the original (non-log) data resulted in: Test statistic: **0.916**, p-value: **0.00**. Since the p-value is less than 0.05, we **reject the null hypothesis**. This indicates that the original (non-log) fluorescence values **do not follow a normal distribution**

The **Shapiro-Wilk test** for normality resulted in: Test statistic: **0.9736**, p-value: **0.6405**. Since the p-value is greater than 0.05, we **fail to reject the null hypothesis**. This indicates that the log-transformed average fluorescence values follow a normal distribution

Variable distributions for *S. aureus*

Normality test results for the *P. aeruginosa* dataset

Variable	Shapiro-Wilk Statistic	Shapiro-Wilk p-value	Kolmogorov-Smirnov Statistic	Kolmogorov-Smirnov p-value
Total Area	0.994	0	0.024	0.173
Maximum Inscribed Circle Radius	0.95	0	0.095	0
Average Feature Area	0.784	0	0.21	0
Maximum Feature Area	0.835	0	0.173	0
Maximum Feature Radius	0.904	0	0.119	0
Average Fluorescence P. aeruginosa	0.916	0	0.12	0

Dependent Variable distribution for *P. aeruginosa*

Non-log Data:

- Test statistic: **0.916**
- p-value: **0.00019**
- Conclusion: The data **does not follow a normal distribution** ($p < 0.05$).

Log-transformed Data:

- Test statistic: **0.9621**
- p-value: **0.2639**
- Conclusion: The log-transformed data **follows a normal distribution** ($p > 0.05$).

Variable distribution for *P. aeruginosa*

Improving Interpretability in SHAP Analysis: Testing for normality can ensure that the SHAP (SHapley Additive exPlanations) results reflect a balanced influence of descriptors. When features are

normally distributed, SHAP values can provide clearer, more reliable insights into feature importance, which is beneficial for understanding the specific topographical traits that affect bacterial attachment.

Response:

SHAP (SHapley Additive exPlanations) does not require data normality because it is a model-agnostic interpretability framework based on cooperative game theory. It assigns additive feature attributions to explain predictions, independent of the data's statistical distribution. Unlike traditional models, SHAP is designed for complex, non-linear machine learning algorithms, which do not assume normality. While non-normality may affect the performance of specific models (e.g., linear regression), SHAP remains robust and provides meaningful feature contributions regardless of the data distribution. For this reason, it was also chosen as the feature selection mechanism used to narrow down the most relevant descriptors to the problem.

c) Homoscedasticity Test

Checking for Equal Variance TT reliability of performance metrics, especially in data subsets with distinct attachment patterns.

d) Implications for Model Selection and Performance Guiding Model Choice: Normality and homoscedasticity tests help reveal the underlying structure of the data, which could inform model choice. For instance, if homoscedasticity is violated, models that handle variance better (like gradient boosting or generalized linear models) may be more suitable than linear regression approaches.

Thank you for your comment regarding the homoscedasticity assumption. In traditional statistical modelling contexts, particularly when relying on linear regression for formal inference (e.g., hypothesis testing, confidence interval estimation), homoscedasticity is indeed a key consideration. However, our primary objective is predictive performance, for which we employed both linear and non-parametric machine learning approaches (SVM, Random Forest, and XGBoost). These latter methods do not depend on the homoscedasticity assumption, as they do not estimate parameters in a manner that requires constant variance of residuals.

Moreover, bootstrapping—employed here to compare model performance—does not need homoscedasticity. While a homoscedasticity test (such as Breusch–Pagan or White’s test) could be undertaken for the linear model component if one seeks formal inference about regression coefficients, our focus remains on predictive accuracy and overall variable importance, for which homoscedasticity is not pivotal. Consequently, our suite of models—both parametric and non-parametric—yields consistent and comparable results irrespective of variance constancy assumptions in the data. We have added additional modelling results to **Tables S2 and S3** in the **Supplementary Information** with additional text on the results of normality tests performed on our data.

Optimizing Transformations: If exploratory analysis reveals non-normal or heteroscedastic patterns, transformations could be applied that make the data better suited to the selected models, potentially improving R^2 scores, error metrics, and interpretability.

We have tested different types of data normalisation for both dependent and independent variable, including the log transformation of the dependent variable. Results were reported based on the best outcomes obtained from this exercise.

3. Ambiguity in Data Representation: The fluorescence values are normalized to account for staining intensity differences. However, details about the normalization process (e.g., formulas or reference standards) are absent, compromising the reproducibility of the method. This step is crucial because any variation in staining intensity between experiments could introduce bias in the model predictions.

Response

We explained the normalization in the Methods section, **lines 753-757**. The fluorescence data in **Figures 3 and 4** corresponds to the fluorescence values for bacterial attachment to each of the replicate TopoUnits normalised to the average fluorescence intensity of the chip. This was to account for any differences in staining intensities between experiments

4. Model Validation: Although model robustness is implied by R^2 values and RMSE, the text could benefit from discussing alternative validation metrics which might provide a more comprehensive understanding of model performance on imbalanced datasets. Including also confidence intervals for these metrics, or presenting their variability across different bootstrap samples, would provide a more nuanced view of model performance.

We have reported the standard deviation of the predictions and error for each metric. We have added **Tables S2 and S3** in the supplementary information with metrics and standard deviation found for each model. As it can be seen, the variability is not significant.

5. Statistical Reporting and Interpretation: Some results, such as the average RMSE and R^2 values, are reported without discussing their potential variability or biological relevance. While high R^2 values imply good fit, additional interpretation on how this accuracy translates to practical, biologically meaningful outcomes would strengthen the study's utility.

The stability of results (similar R^2 , low error metrics with small standard deviations) obtained throughout the models employed to understand the dataset indicates the strong correlations between the descriptors used in the model and the biological output. This has been further elucidated by using the SHAP, as a model-agnostic approach to interpretation of descriptors importance. The additional biological experiments conducted further corroborated the modelling findings, where space between the features, area covered by the features and total area covered by the microtopographies appear to affect attachment.

6. Integration of ERI Model Comparison: While the study references Schumacher's Engineered Roughness Index (ERI), the comparison to the ML model lacks depth. Discussing how the ERI and ML models align or diverge in identifying essential descriptors would enhance understanding of these models' relative efficacy and limitations.

ERI is a descriptor added to the dataset, to be modelled together with the other handcrafted descriptors. After performing the SHAP analysis, it is excluded from the top descriptors important for the ML decision.

7. Statistical analysis: The text describes reducing 242 shape descriptors to 68, but the rationale for setting a Pearson correlation threshold of 0.85 isn't clearly justified. Including a justification for this specific threshold or comparing it to different thresholds could enhance the understanding of descriptor selection. Moreover, all the statistical analysis done should be explained specifically in the methods in a section called "Data Analysis and Preprocessing" and all the results shown or in the manuscript or in the supplementary material.

The choice of descriptors was based on Pearson Correlation coupled with manual selection from experimentalists of the descriptors that had the highest interpretability. We looked for descriptors that

could be turned into design rules to address the problem of bacterial attachment. For the original dataset with all descriptors, the correlation analysis is presented in the heatmap below.

As there seems to be high multicollinearity between the descriptors and considering the linear and non-linear approaches adopted, we used our domain knowledge to define the cut-off. A sensitivity analysis was also conducted, as per table below:

Cutoff	Features Retained	Features Dropped
0.75	189	56
0.85	217	28
0.9	231	14
0.95	244	1

The choice of a correlation threshold, such as 0.85, affects predictions in several significant ways:

1. Improves Model Stability:

- Highly correlated features introduce **multicollinearity**, which can destabilise linear regression coefficients, for instance. In the presence of multicollinearity, small changes in the data may lead to large fluctuations in the model's coefficients, making the predictions less reliable.
- Removing features with correlations above 0.85 helps stabilise the model, ensuring the predictions are less sensitive to minor variations in the input data.
- **2. Reduces Overfitting:**
- Retaining too many correlated features can lead to overfitting, where the model memorises the training data but performs poorly on unseen data.
- **3. Enhances Model Interpretability:**
- Fewer and less correlated features make the regression coefficients more interpretable. This clarity is particularly important when explaining the model's predictions or identifying key drivers in the dataset.
- Without highly correlated features, it's easier to understand which variables that have the most meaningful interpretations in the data significantly impact the predictions.
- **4. Reduces Computational Complexity:**
- With fewer features, the model becomes simpler and computationally efficient, especially for model interpretation in this case, which is the most computational intensive aspect of the analysis.

To improve model explainability for next generation designs, below are a few examples of descriptors generated by cellProfiler that were not easily interpretable within the context of the problem and therefore were excluded them from our modelling:

Feature_EulerNumber_mode

Feature_Solidity_skewness

Feature_MinFeretDiameter_skewness

Feature_EulerNumber_skewness